

# Simulation of the transport, vertical distribution, optical properties and radiative impact of smoke aerosols with the ALADIN regional climate model during the ORACLES-2016 and LASIC experiments.

Marc Mallet[1], Pierre Nabat[1], Paquita Zuidema[2], Jens Redemann[3], Andrew Mark Sayer[4,5], Martin Stengel[6], Sebastian Schmidt[7], Sabrina Cochrane[7], Sharon Burton[8], Richard Ferrare[8], Kerry Meyer[5], Pablo Saide[9], Hiren Jethva[4,5], Omar Torres[5], Robert Wood[10], David Saint Martin[1], Romain Roehrig[1], Christina Hsu[5] and Paola Formenti[11]

Affiliation

[1] Centre National de Recherches Météorologiques, UMR3589, Météo-France-CNRS, Toulouse, France

[2] Rosenstiel School of Marine and Atmospheric Sciences, University of Miami, Miami, FL, USA

[3] University of Oklahoma, USA

[4] Universities Space Research Association, Columbia, MD, USA

[5] NASA Goddard Space Flight Center, Greenbelt, MD, USA

[6] Deutscher Wetterdienst (DWD), Offenbach, Germay

[7] University of Colorado, Boulder, USA

[8] NASA Langley Research Center, Hampton, Virginia, USA

[9] University of California, Los Angeles (UCLA), USA

[10] University of Washington, Seattle, USA

[11] Laboratoire Interuniversitaire des Systèmes Atmosphériques, UMR CNRS 7583, Université Paris Est Créteil et Université Paris Diderot, Institut Pierre Simon Laplace, France

*Correspondence to*: M. Mallet (marc.mallet@meteo.fr)





**Abstract**

Estimates of the direct radiative forcing (DRF) from absorbing smoke aerosols over the Southeast
Atlantic Ocean (SAO) requires simulation of the microphysical and optical properties of
stratocumulus clouds (Sc) as well as of the altitude and shortwave (SW) optical properties of
biomass burning aerosols (BBA). In this study, we take advantage of the large number of
observations acquired during the ORACLES-2016 and LASIC projects during September 2016 and
compare them with datasets from the ALADIN-Climate regional model. The model provides a good
representation of the liquid water path (LWP) but the low cloud fraction (LCF) is underestimated
compared to satellite data. The modeled total column smoke aerosol optical depth (AOD) and
Above Cloud AOD (ACAOD) are consistent (~0.7 over continental sources and ~0.3 over SAO at
550 nm) with MERRA2, OMI or MODIS data. The simulations indicate smoke transport over SAO
occurs mainly between 2 and 4 km, consistent with surface and aircraft lidar observations. The BBA
single scattering albedo (SSA) is slightly overestimated compared to AERONET, and more
significantly when compared to Ascension Island surface observations. The difference could be due
to the absence of internal mixing treatment in the ALADIN-Climate model. The SSA overestimate
leads to underestimate the simulated SW radiative heating compared to ORACLES data. For
September 2016, ALADIN-Climate simulates a positive (monthly mean) SW DRF of about +6
W.m$^{-2}$ over SAO (20°S–10°N and 10°W–20°E) at the top of the atmosphere (TOA) and in all-sky
conditions. Over the continent, the presence of BBA is shown to significantly decrease the net
surface SW flux, through direct and semi-direct effects, which is compensated by a decrease
(monthly mean) in sensible heat fluxes (-25 W/m$^{-2}$) and surface land temperature (-1.5 °C) over
Angola, Zambia and Congo notably. The surface cooling and the lower tropospheric heating tends
to decrease the continental planetary boundary layer (PBL) height by about ~200 m.

**1. Introduction**

Southern Africa is one of the main sources of biomass burning aerosols (BBA) at the global scale.
When the intense smoke plumes are transported over the Southeast Atlantic Ocean (SAO), they are
able to produce a significant positive direct radiative forcing (DRF) at the top of the atmosphere
(TOA) in the SW spectral range and in all-sky conditions (De Graaf et al., 2012, 2014, Feng and
Christopher, 2015, Zuidema et al., 2016). Over this specific region, the sign of the DRF is found to
be opposite to the « well-known » cooling effect generally exerted by natural and anthropogenic
aerosols at TOA. Based on the combination of satellite observations from A-Train data sets
(MODIS, CERES and OMI), Feng and Christopher (2015) indicate a regional-averaged



instantaneous (i.e., time of observations) DRF of about +37 W.m$^{-2}$ (regional mean; 20°S–10°N and
     10°W–20°E) for August 2006, with the highest magnitude of the forcing reaching +138 W.m$^{-2}$ at
     TOA. Significant positive values are also underlined by De Graaf et al. (2012, 2014), who estimate
     an averaged DRF (August 2006) of about +23 W.m$^{-2}$ near the Southern African coast. In parallel,
     Meyer et al. (2013) report an instantaneous (near local noon for Aqua overpass) regional mean

above-cloud radiative forcing efficiency from 50 W.m$^{-2}$AOD$^{-1}$ to 65 W.m$^{-2}$AOD$^{-1}$ by using their
     bias-adjusted MODIS cloud retrievals. By using SCIAMACHY observations and radiative transfer
     model calculations, De Graaf et al. (2014) further estimate a SW DRF of about ~+ 30/35 W.m$^{-2}$ over
     the same domain in August and September (2006-2009).

     This positive sign of the DRF is mainly due to the presence of highly reflective stratocumulus

clouds (Sc) over the SAO. Although such a positive DRF is occasionally observed over other
     regions, such as the northeast Pacific during extreme summertime biomass burning events in
     continental North America (Mallet et al., 2017), the SAO clearly represents the main region at the
     global scale where such positive forcings can be observed every years at a seasonal-time scale.
     Indeed, this significant radiative forcing is due to the persistent biomass burning emissions over

Central Africa during the July-August-September-October (JASO) period. Smoke emissions over
     the central Africa are also related to a significant inter-annual variability, associated with an
     important increase over the 1979-2015 period (Hodnebrog et al., 2016).

     All studies clearly underline the importance of both the aerosol radiative properties of smoke
     plumes (e.g., Aerosol Optical Depth, AOD, single scattering albedo, SSA), their vertical structures

(notably, the localization of smoke vs. Sc; Johnson et al., 2004) as well as the underlying cloud
     properties (e.g., cloud optical depth (COD), liquid water path (LWP)) on the produced positive SW
     DRF at TOA. As an example, Feng and Christopher (2015) report a critical COD of ~12-20 capable
     of changing the sign of the DRF from negative to positive (at TOA) for BBA characterized by SSA
     ~0.91 and AOD ~1.0 (at 550 nm). In the case of more absorbing smoke (SSA ~0.85), the ranges for

critical COD are strongly reduced and reach ~2-4. Chand et al. (2009) also underline the importance
     of cloud coverage on the DRF exerted at TOA by smoke over SAO. In addition, Sakaeda et al.
     (2011) provided model estimates of regional radiative forcing from direct and semi-direct effects,
     with important implications on cloud properties (cloud fraction notably). These complex processes,
     involving both microphysical and optical properties of BBA and Sc explain, at least partially, the

large difficulty of recent global climate models (GCM) in reproducing the DRF of smoke over this
     specific region (Stier et al., 2013).

     In that context, it appears crucial to evaluate carefully and constrain both smoke aerosols and Sc
     properties in GCM or in their regional configurations (regional climate models, RCM) before




running them over a long-time period, for radiative budget and climatic considerations. The main

objectives of this study consist to investigate the transport of BBA over SOA, the vertical layering as well as optical properties using the ALADIN-Climate model. In addition, the induced SW DRF at TOA and the possible impact of BBA on the regional (continental) climate are also analysed. This work has been conducted in the context of several international field campaigns over the SAO region, including the ObseRvations of Aerosols above Clouds and their intEractionS (ORACLES)

(Zuidema et al., 2016), the Layered Atlantic Smoke Interactions with Clouds (LASIC, Zuidema et al., 2018), the AErosol RAdiation and CLOuds in Southern Africa (AEROCLO-sA) and the Cloud-Aerosol-Radiation Interactions and Forcing : Year 2017 (CLARIFY) projects. More specifically, this study takes advantage of the large number of in situ observations acquired from aircraft and surface measurements during September 2016 for the ORACLES-1 and LASIC projects. This

unique dataset is combined to satellite aerosol and cloud retrievals (MODIS, OMI and SEVIRI) and re-analysis (MERRA-2 and MACC) data.

We focus our analyses on specific properties which are important to study the radiative effect of BBA over SAO. For Sc clouds, these are low cloud fraction (LCF), liquid water path (LWP) and COD. For BBA, special attention is paid to AOD, Above Cloud AOD (ACAOD), extinction vertical

profiles, SSA, and SW radiative heating due to smoke. The DRF at the surface and TOA are estimated and analysed in addition to climatic implications, especially those exerted by BBA over the Central Africa continent. The regional modeling model used in this work is the ALADIN-Climate model (Nabat et al., 2015a, 2015b; Daniel et al., 2018), which has been modified in a recent configuration to better represent smoke aerosols and notably their SW optical properties.

This article is organized as follows. First, details on these recent developments are provided in section 2 as well as the design of the ALADIN-Climate simulations. Section 3 reports the complete dataset (satellites, reanalysis, in situ surface and aircraft observations). The analyses of the comparisons between simulated and observed Sc and aerosol properties are presented in section 4. Based on the comparisons, we analyse more specifically the concentration of smoke aerosols over

biomass burning sources and during the transport, the altitude of BBA, as well as absorbing properties and induced SW heating rate due to smoke. In addition, the impact of the elevated relative humidy within smoke plumes on optical properties is also investigated in section 4. Finally, section 5 focuses on the analyses of the SW DRF exerted by smoke aerosols at TOA during September 2016, as well as their impact on the continental climate in terms of the surface energy

budget (temperature, sensible heat fluxes) and lower troposphere dynamics (planetary boundary layer (PBL), notably).

## 2. The regional ALADIN-Climate model



### 2.1. Aerosol Scheme

The recent aerosol scheme (TACTIC, Tropospheric Aerosols for ClimaTe in CNRM-CM) included
in the ALADIN-Climate model accounts for sulfate, organic (OC) and black (BC) carbon, dust and
primary sea-salt particles (Nabat et al., 2015b; Michou et al., 2015). This model includes advection
by atmospheric winds, diffusion by turbulence, surface emissions as well as dry and wet (in-cloud
and below cloud) removal processes. In ALADIN-Climate, mineral dust and sea-salt emissions are
interactively connected with surface meteorological fields and soil properties (Nabat et al., 2015a).
For the primary BC and OC species and secondary sulfates, a bulk approach is applied whereby a
fixed aerosol size distribution is assumed for calculating aerosol properties, while for mineral dust
and sea salt particles, a more explicit size representation is used based on 3 bins for dust and sea-
salt. The TACTIC scheme assumes an external mixture of the different aerosol species. For specific
situations, this could potentially represent a limitation, especially with regard to possible BC mixing
(internal/external) state, which can significantly affect SW absorption (Fierce et al., 2016) by
aerosols. Knowing that, specific attention is being paid in this study to the simulated absorbing
properties (SSA) of BBA, as well as the associated SW heating rate.

The radiative properties (mass extinction efficiency (MEE), SSA, and asymmetry parameter (ASY))
of each aerosol species are calculated for the different spectral bands of the Fouquart and Morcrette
radiation scheme (FMR, Morcrette, 1989) and the rapid radiative transfer model (RRTM), for the
SW and Longwave (LW) radiations, respectively. Aerosol DRF at the surface and at TOA (in SW
and LW spectral range and for both clear-sky and all-sky conditions) is diagnosed using a double
call (with and without aerosols) to the radiation schemes during the model integration. In addition,
the semi-direct radiative forcing, which represents the modifications of the cloud properties and
atmospheric dynamics due to absorption of SW radiations by smoke, is also represented. In its
current version, BBA are represented by two different tracers (primary BC and OC) with fixed
microphysical and radiative properties without any consideration of possible differences between
fossil fuel and biomass-burning emissions. This hypothesis implies that the radiative, hygroscopic
properties and e-folding time (aging) of carbonaceous species are similar for both anthropogenic
and smoke emissions.

### 2.2 Smoke Radiative properties

Two tracers have been recently implemented in ALADIN-Climate describing, respectively, the mass
concentration of fresh and aged smoke aerosols, following the methodology presented in Bellouin et
al. (2011). Aging from the fresh mode to hygroscopic aged is quantified using an e-folding time of 6
hours according to Abel et al. (2003). This value is two times higher than the one (~3 h) recently
proposed by Vakkari et al. (2018) for southern African savannah. The smoke over the SAO will be



more aged, on the order of 5-7 days (Adebiyi and Zuidema, 2017; Diamond et al., 2018). While studies of the BBA chemical composition and attribution for the smoke's optical and hygroscopic properties are still ongoing, preliminary results indicate further smoke aging increases its ability to

function both as a cloud condensation nucleus and to absorb shortwave radiation (Zuidema et al., 2018).

For each tracer, dry-state aerosol size distributions are assumed based on lognormal function (Table 1) similar to those implemented in the Hadley Centre global climate model, HadGEM2-ES (Bellouin et al., 2011). The smoke dry-state refractive indices used to calculate radiative properties

are also reported in the Table 1 (at 550 nm). The values of the real and imaginary refractive indices have been updated using the AERONET observations obtained by Eck et al. (2013) in Zambia (Mongu station). Although they indicate a pronounced seasonal cycle in the real and imaginary parts of the refractive index from AERONET data, we have used a mean value of 0.03 (at 550 nm) for the imaginary component in our Mie calculations (Table 1). SW radiative properties have been

calculated for the specific wavelength bands of the FMR radiation scheme. The values in the SW spectral ranges are reported in the Table 1. At 550 nm and in dry state, the calculated radiative properties are 5.0 $m^2.g^{-1}$, 0.81 and 0.67 for the MEE, SSA, and g for the « fresh » smoke tracer (Table 1). The values for « aged » smoke are, respectively, 6.0 $m^2.g^{-1}$, 0.90 and 0.80 (Table 1). As BBA are known to be hydrophilic (Rissler et al., 2006), the dependence of the radiative properties

to relative humidity (RH) has been included for both tracers. This dependence is formulated as described by Solmon et al. (2006) :

$$MEE_{wet}= MEE_{dry}(1-RH)^{-\alpha} \tag{1}$$

where $MEE_{wet}$ and $MEE_{dry}$ are for wet and dry conditions. We have selected a value of 0.26 and 0.15 for the parameter α in order to reproduce the changes of MEE with RH for aged and fresh

smoke, respectively. At very high humidity (RH > 99%) maximum thresholds of 8.5 and 16.9 $m^2.g^{-1}$ are considered for fresh and aged smoke, in order to avoid unrealistic values of MEE. In a similar way, we have also implemented a dependence of smoke SSA on RH using the same relationship as (1) (Mallet et al., 2017). The values of α have been fixed to 0.015 (0.02) for aged (fresh) smoke to represent the variations of SSA with RH, as reported in Bellouin et al. (2011).

**2.3 Aerosols and Cloud interactions**

"Aerosol-cloud" interactions were represented using a simple parametrization, similar to most GCMs, thereby maintaining the low numerical costs necessary for climate and ensemble simulations. The activation of hydrophilic particles to cloud droplets is not explicitly resolved in ALADIN-Climate and the first indirect radiative effect is implemented for hydrophilic sulfates,

organic carbonaceous and sea-spray aerosols. This first indirect effect is represented by a simple



relationship in ALADIN-Climate relating the mass of hydrophilic aerosols to the cloud droplet number concentration (CDNC) based on the work of Martin et al. (1994). The radiative properties (COD, SSA and ASY) of liquid clouds are calculated in the SW spectral region by the parameterizations proposed by Slingo and Schrecker (1982). In the present work, we do not discuss

possible first indirect effects between BBA and Sc which will be addressed and analyzed in a specific future companion study. The impact of aerosols on liquid clouds via the second indirect effect (precipitation modulation due to the hygroscopic aerosols) is currently under development. In these simulations, the autoconversion rate from water cloud to rain is not sensitive to the aerosol loading and the value of $8.10^{-4}$ kg.kg$^{-1}$ (Smith et al., 1990) is used for the critical cloud water

mixing ratio.

**3. Model Configuration and Data Used**

**3.1. Simulation Design and Important Physics Options**

The ALADIN-Climate simulations cover the period from 01 August to 31 October 2016. The lateral boundary conditions are provided by ERA-INTERIM (Dee et al., 2011). The possible long-range

transport of BBA is not forced at the lateral boundary conditions but rather a large domain is defined encompassing the main biomass-burning sources. The horizontal resolution of the model is 12 km with 91 vertical levels (from 1015 to 0.01 hPa). The land surface is treated using the SURFEX model (Masson et al., 2013). As detailed latter, we also use a spectral nudging method described in Radu et al. (2008). The FMR (RRTM, Mlawer et al., 1997) radiative transfer scheme is

used to calculate the SW (LW) radiation. Finally, it should be mentioned that the possible impact of BBA reducing the sea-surface temperature (SST) is not treated here and the ALADIN-Climate model is used in a forced mode configuration. This possible impact of BBA is outside the scope of the present study.

The biomass-burning emissions from the CMIP6 inventory have been used for BC, OC and sulfur

gaseous SO2. Following the study of Petrenko et al. (2017), an adjustment factor of 2.5 is applied to the biomass burning emissions. BBA are emitted into the first vertical level of the model, without any considerations of pyroconvective processes, as no clear consensus on such processes exists over this region. For example, Labonne et al. (2007) showed that smoke plumes are generally confined in the planetary boundary layer (PBL) near smoke sources. Accordingly, smoke emissions force the

model at the first model level following the recommendations from the first phase of AEROCOM (Dentener et al., 2006). Fire emissions from the savannah are emitted at the lowest model level, allowing subgrid-scale turbulence mixing through the boundary layer. The diurnal cycle of smoke emission is not taken into account, which could impact the temporal variations of the aerosol loadings (Xu et al., 2016). We assume that the main smoke emissions transported over SAO are



included in the domain defined in Figure 1. Finally, a climatology is used in the model for organic

aerosols produce from vegetation biogenic emission.

The BBA mass is known to increase during aging due to the condensation of volatile organic

compounds. In the absence of a explicit representation of secondary organic aerosols (SOA)

production in ALADIN-Climate, a ratio of particulate organic matter (POM) to primary OC has

been used for artificially representing SOA formation within the smoke plume. The lack of a

complete representation of SOA in current climate models obviously represents an important source

of uncertainties in the estimation of BBA concentration (Johnson et al., 2016). For this ALADIN-

Climate simulation, an average POM/OC ratio of 2.3 is applied (Formenti et al., 2003) based on

SAFARI-2000 data. This value is consistent with the recent results obtained by Vakkari et al. (2018)

and higher than the one (1.6) retained in the HadGem global model (Bellouin et al., 2011, Johnson

et al., 2016).

Two sensitivity tests performed using different POM to OC ratios (2 and 3) have been performed,

showing an impact of ±0.15 on BBA AOD over the continent (Figure S1 in the supplement

material). An additional simulation tested the sensitivity of BBA AOD to the e-folding time using

the recent value proposed by Vakkari et al. (2018). The results (Figure S1 in Supplement Material)

indicate a slight AOD decrease of about -0.05 when averaged over the box_S (15-25°E / 5-15°S, see

Figure 1).

Three ALADIN-Climate simulations (excluding the sensitivity tests only shown in Figure S1,

Supplement Material) are performed. The first one (control run, CTL) does not take BBA into

account, while the second simulation (named SMK) includes the direct and semi-direct radiative

effect of BBA. Finally, a nudged (on wind and relative humidity) simulation (named SMK_SN)

investigates more specifically the impact of the water vapor transported within the smoke plume on

BBA optical properties and the associated SW radiative heating. For the latter, the nudging does not

affect PBL, which can be independently influenced by the radiative effects of smoke.

**3.2 Surface, Aircraft, Satellite and Reanalysis dataset**

Different datasets of aerosol and cloud properties from surface, remote-sensing, and reanalysis,

have been used for evaluating the ALADIN-Climat simulations. Satellite and reanalysis data are

summarized in the Table 2.

**3.2.1. LASIC Surface Observations (Ascension Island)**

SSA (at 529 nm) at Ascension Island was estimated from the in-situ measurement of the scattering

coefficient estimated by a nephelometer and the absorption coefficient deduced from a Particle Soot

Absorption Photometer (PSAP). The PSAP measurements incorporate an average of the Virkkula

(2010) and Ogren (2010) wavelength-averaged corrections, and are collected at standard





temperature and pressure, with dilution corrections applied. The RH of the air entering the PSAP is estimated to be 25% or less, while the air entering the nephelometer is measured, with values ranging between 45%-60%. Differences in the RH are speculated to bias the SSA higher rather than lower, because drying will reduce the coating thickness on the refractory black carbon, reducing lens-induced enhancement of shortwave absorption. The original nephelometer scattering measurements at 550 nm are converted to estimated values at 529 nm using the scattering-derived

angstrom exponent. These measurements are also reported in Zuidema et al. (2018). An independent evaluation of the SSA in August-September 2017 using an Aerodyne Cavity Attenuated Phase Shift-SSA instrument is consistent with the values reported here (Tim Onasch, personal communication).

**3.2.2. AERONET retrievals**

Two continental (Mongu and Lubango) and one maritime (Ascension Island) AERONET sites

extend local comparisons to the atmospheric column and for different aerosol variables. As described by Dubovik and King (2000), the AERONET network allows retrieval of microphysical (volume size distribution) and optical (refractive indexes, SSA, ASY and scattering/absorption optical depth) properties of aerosols, as well as their spectral dependence in SW spectral range. The uncertainty of retrieved SSA is ± 0.03 for AOD (440 nm) > 0.2 for water soluble aerosols and for

AOD (440 nm) > 0.5 (zenith angle larger than 50 degrees) for desert dust and BBA. For AOD (440 nm) < 0.2, the SSA accuracy is ± 0.05–0.07 (Dubovik et al., 2000). We focus our analyses on Level 2 AOD and SSA.

**3.2.3 Aircraft Observations**

3.2.3.1 Aerosol Extinction profiles

The NASA Langley 2nd generation High Spectral Resolution Lidar (HSRL-2) has been in operation during ORACLES-1 aboard the NASA ER2. HSRL-2 measures particulate backscatter and extinction at 355 nm and 532 nm using the HSRL technique (Shipley et al., 1983) and aerosol backscatter at 1064 nm. All three wavelengths also measure depolarization. The HSRL technique uses a separate filtered channel at each HSRL wavelength that is sensitive to molecular scattering,

but not aerosol scattering, and which therefore provides with a direct observation of the attenuation of the signal. From this, the vertically-resolved particulate extinction is derived by comparison to a molecular density profile from direct measurement or a model. For ORACLES-1, the HSRL-2 retrieval uses molecular density profiles from MERRA-2 (Gelaro et al. 2017). The filtering is accomplished with an iodine gas filter at 532 nm (Hair et al. 2008) and a density-tuned field-

widened Michelson interferometer at 355 nm (Burton et al. 2018). More information about the instrument, calibrations, and algorithms is given by Hair et al. (2008) and Burton et al. (2015, 2018). The vertical resolution for extinction is 315 m and for backscatter and depolarization is 15





m. The horizontal resolution is 60 seconds for extinction and 10 seconds for backscatter and depolarization, or approximately 10 km (extinction) and 1.8 km (backscatter). The ORACLES

HSRL-2 extinction product can be found at https://espoarchive.nasa.gov/archive/browse/oracles/id8/ER2.

3.2.3.2 SW Heating Rate estimates

Heating rate profiles segregated by absorber (aerosols, water vapor, oxygen)are determined using the spectral information from the Solar Spectral Flux Radiometer (SSFR). SSFR measures

upwelling (nadir) and downwelling (zenith) irradiance from 350-2100 nm. The zenith light collector is actively leveled, which allows SSFR to obtain spectral irradiance measurements throughout spiral profiles that extend from the top of the aerosol layer to the bottom of the cloud layer. These measurements lend themselves to a new algorithm for retrieving aerosol SSA and ASY from 350-860 nm. It uses measurements made during the spiral aircraft descents to separate changes in

upwelling, downwelling and net irradiance due to the aerosol layer from those due to the underlying cloud field (Cochrane et al., 2018).

The 4STAR spectral AOD, HSRL-2 extinction profiles, and the spectral SSA and ASY retrievals, provide the inputs for the heating rate profiles calculated with the LibRadtran radiative transfer tool (Mayer and Kylling, 2005). The retrieved intensive aerosol properties SSA and ASY are vertically

homogeneous, whereas the spectral extinction coefficient from the merged 4STAR HSRL-2 measurements varies with altitude. The cloud albedo below the aerosol layer is directly measured by SSFR, and the atmospheric water vapor profile is determined from in situ measurements. Using these inputs, two calculations of the heating rate profile are done: one with, and one without aerosols, and the difference is reported as aerosol heating rate. The accuracy of the calculation is

ensured by comparing the calculated irradiance spectrum above and below the aerosol layer with the SSFR measurements. After this step, the heating rate is spectrally integrated over the solar wavelength range (380-2125 nm).

**3.2.4 Satellite Retrievals**

3.2.4.1 MODIS and MISR dataset

The Deep Blue ACAOD retrieved from the MODIS instrument are based on a slightly updated version of the demonstration algorithm presented in Sayer et al. (2016). In brief, this algorithm performs a multispectral weighted least-squares fit of measured reflectance in four bands across the visible spectral region (centered near 470, 550, 650, and 870 nm) to simultaneously retrieve ACAOD and the COD. The minimization is performed using optimal estimation. This provides with

estimates of the uncertainty on retrieved parameters, as well as an indicator of how well the retrieval solution is able to fit the measurements. Retrievals where the forward model is expected to



be inappropriate, or where the measurements do not constrain the retrieved quantities, are filtered out.

The main updates since Sayer et al (2016) are twofold. First, the radiative transfer lookup tables
have been updated to include dimensions for surface pressure and surface albedo, which improves the realism of the forward model over land. Surface pressure is estimated using terrain altitude and an assumed scale height of 7.4 km, while the surface albedo is taken from a climatology based on the MODIS gap-filled snow-free land albedo data set (Sun et al., 2017). The second is that, rather than applying the retrieval to each individual pixel, the pixels are aggregated into 10 km x10 km
(effective nadir resolution) boxes, chosen to match the resolution of the level 2 MODIS aerosol products. Then, the median reflectance of water cloud pixels within each box is used for the retrieval. Use of median reflectance decreases sensitivity to factors such as 3D effects or cloud detection errors. Additionally, if a 10 km x 10 km pixel has a water cloud fraction under 0.75, it is excluded, for the same reason.

The MOD06ACAERO (Meyer et al., 2015) products are also used. These use reflectance observations at 6 MODIS spectral channels (0.46, 0.55, 0.66, 0.86, 1.24, and 2.1 μm) to simultaneously retrieve ACAOD, and the COD and CER of the underlying marine boundary layer clouds. Retrievals are performed at the pixel level (here, every fifth native 1 km pixel) on both Terra (morning) and Aqua (afternoon) MODIS data. Output includes pixel-level estimates of retrieval
uncertainty that accounts for known and quantifiable error sources (e.g., radiometry, atmospheric profiles, cloud and aerosol radiative models). Assumptions regarding the cloud forward model and ancillary data usage are consistent with those of the operational MODIS cloud products (MOD/MYD06) (Platnick et al., 2017). Note that both these data sets represent only the partial column AOD, i.e. the AOD above the liquid cloud top, and that ice phase clouds are not processed.

In addition, we have also used  MODIS-Terra and Aqua combined Deep Blue/Dark Target data set (AOD_550_Dark_Target_Deep_Blue_Combined_Mean_Mean) from the latest Collection 6.1 (Sayer et al., 2014) and MISR (MIL3MAE monthly mean data at 0.5° of resolution; Kahn et al., 2015) (see Table 2).

3.2.4.2 OMI dataset

The Ozone Monitoring Instrument (OMI) sensor, operating since October 2004 onboard of the EOS Aura satellite, is a spectrometer with a high spectral resolution (Levelt et al., 2006). OMI offers nearly the daily global coverage with a spatial resolution for the UV-2 and VIS (UV-1) channels ranging from $13 \times 24$ km$^2$ at nadir. The OMAERUV_v003 product contains retrievals from the OMI near-UV algorithm (Torres et al., 2007). This algorithm derives a variety of aerosol radiative
properties, such as an aerosol index (AI), AOD, AAOD (uncertainty of ± (0.05 + 30%)) for clear-



sky conditions. For this study, we have used ACAOD (Jethva et al., 2018) retrieved at 500 nm (Table 2). An above-cloud aerosol retrieval technique was also applied to the multi-year record of OMI observations to deduce a global product of ACAOD on a daily scale (Jethva et al., 2018).

3.2.4.3. SEVIRI dataset

Spatiotemporally highly resolved geostationary satellite observations are taken here from the CLoud property dAtAset based on SEVIRI edition 2 (CLAAS-2, Benas et al., 2017). The CLAAS-2 dataset is based on measurements of the Spinning Enhanced Visible and Infrared Imager (SEVIRI) and was generated and released by the EUMETSAT Satellite Application Facility on Climate Monitoring (CM SAF) as the successor of CLAAS (Stengel et al., 2014). CLAAS-2 includes a

variety of cloud properties, of which LWP, COD and CER were used in this study. CLAAS-2 COD and CER are retrieved, similarly to the widely used cloud retrieval method described in Nakajima and King (1990), under the assumption of plane-parallel cloud layers. Lookup tables are pre-calculated and used to map SEVIRI reflectance at 0.6 μm and 1.6 μm wavelengths to COD and CER as function of satellite-sun geometries and cloud phase. For liquid clouds, COD and CER are

used to calculate LWP following Stephens (1978). The algorithm is initially described in Roebeling et al. (2006) with more details and updates given in Benas et al. (2017), which also report validation exercises. The CLAAS-2 Level-2 data are instantaneous data on native SEVIRI resolution with a temporal resolution of 15 minutes. For this study, the data are projected onto a regular latitude-longitude grid using nearest neighbor approach.

**3.2.5. Monitoring Atmospheric Composition and Climate and Modern-Era Retrospective Analysis for Research and Applications-II Aerosol and Clouds Reanalysis**

Two different reanalyses are used for aerosols and clouds (Table 2). The European Center for Medium-Range Weather Forecasts (ECMWF) reanalysis of global atmospheric composition since 2003 includes five main aerosol species. The first generation of ECMWF reanalysis (Morcrette et

al., 2009), issued by the GEMS (Global and regional Earth-System (atmosphere) Monitoring using Satellite and in situ data) project, covers the period 2003-2008. The Monitoring Atmospheric Composition and Climate (MACC) is the second-generation product and provides improvements in sulfate distributions and has extended to the 2003-2011 period (Benedetti et al., 2009). Here, we use MACC NRT daily data sets at 1.125° resolution of the anthropogenic SW direct forcing at TOA in

all-sky conditions (Table 2). In addition, we use the Modern-Era Retrospective Analysis for Research and Applications (MERRA) reanalysis, generated with version 5.2.0 of the Goddard Earth Observing System (GEOS) atmospheric model and data assimilation system (DAS). The system, the input data streams and their sources, and the observation and background error statistics are



fully documented in Rienecker et al. (2011). We rely on the AOD for the different species at 0.5° ×
0.625° spatial resolution (Table 2).

**4. Microphysical and Optical properties of Sc and BBA**

**4.1. Sc properties**

The different properties of Sc are analysed over the box 10-20°S / 0-10°E, defined by Klein and
Hartmann (1993), referenced in the following as box_O.

**4.1.1. Macrophysical and Microphysical Sc properties**

Figure 2a reports the daily mean LWP (g.m$^{-2}$), LCF as well as liquid COD, averaged over the box_O
for September 2016. A reasonable agreement is evident in the LWP between ALADIN-Climate
(SMK simulation), SEVIRI and ERA-INT data. The simulated LWP is in the range of ERA-INT
and SEVIRI data, if slightly less, by -8 g.m$^{-2}$ in the mean compared to ERA-INT. The LWP maxima
are also well simulated, especially for the 25-28 September, with LWP ~90 g.m$^{-2}$. The temporal
correlation is about ~0.45 between ERA-INT and the SEVIRI LWP.

In contrast, an important negative bias of approximately ~-20% is detected in the simulated low
cloud fraction compared to ERA-INT values (Fig. 2b). This result is related to the well-known « too
few, too bright» bias detected in most GCMs over the Sc regions (Nam et al., 2012). The mean
modeled LCF is 57%, compared to 80% in the ERA-INT dataset. The poor representation of LCF
over Sc regions remains an open issue outside of the scope of this work. Nevertheless, the analysis
and discussions of the following results take into account this underestimates, notably for SW
heating rate and the DRF exerted by BBA at TOA.

**4.1.2. Optical Sc properties**

Figure 2c represents the daily COD estimated by ALADIN-Climate using the Slingo et al. (1998)
and Nielsen et al. (2015) parameterizations. This figure also includes an ALADIN-Climate
simulation with a fixed CER of 10 μm, close to the SEVIRI values (Figure S2 in Supplemental
Material). Retrieved values from MODIS and SEVIRI are also reported in Figure 2c. The SEVIRI
and MODIS-Aqua values are consistent with each other, for a mean COD of ~8 for both
instruments. The COD is overestimated by the model in both configurations but especially when the
Slingo parameterization is used (average COD value of 11.5). The Nielsen parametrization slightly
reduces the bias for a mean COD of 9.5. As the LWP is realistically simulated by the model, this
negative bias could be due to errors (underestimates) in the simulated CER by ALADIN-Climate.
Our sensitivity test conducted using a fixed CER of 10 μm indicates a reduced bias (+0.25). As
mentioned previously, the BBA indirect radiative effect is not addressed in this study but is known
to be an important issue over SAO (Costantino and Bréon, 2013, Lu et al., 2017). The impact of
BBA on cloud microphysical properties will be studied in a future work.



### 4.2. Biomass Burning Aerosols

### 4.2.1. AOD over biomass burning source

Figure 3 compares the (monthly mean) total (clear-sky, not above-cloud) AOD simulated by ALADIN-Climate to those derived from MODIS-Terra and Aqua combined Deep Blue/Dark Target data set (AOD_550_Dark_Target_Deep_Blue_Combined_Mean_Mean) from the latest Collection 6.1 (Sayer et al., 2014) and MISR. The total AOD obtained from the MERRA-2 reanalyses is also indicated. All AODs are at 550 nm. Over Angola and Zambia, the model is able to simulate a

regional pattern of AOD consistent with the MODIS and MISR retrievals, with AODs of ~0.7-0.9, even if BBA are located too far south compared to satellite data. Over the continent, significant differences (overestimates) appear clearly compared to MERRA2. This difference with reanalyses product can be due to the scaling factors applied for biomass burning emissions in the different models. Over the ocean, ALADIN-Climate is found to be very consistent with MERRA2 (AOD

~0.7 near the Angola coast) but the differences are large compared to the satellite retrievals. The latter AODs are higher (~0.7-1.0), especially for MODIS. It should be mentioned that 50% of MODIS AOD over the SAO is due to coarse mode according the level 2 retrievals.

In addition, the land-ocean contrast in AOD detected by MODIS and MISR, with lower AODs over the continent and higher AODs over the ocean, is not observed in ALADIN-Climate. This contrast

in AOD is still detected in MERRA2 data, but it is not as large as in the satellite retrievals. It is likely that some of the land/ocean contrast in the satellite data comes from two factors. The first is that the over-land and over-water algorithms are different and may have different biases. The second is that cloud fraction is also significantly higher over the water than over the land, meaning that typically more days of data contribute to the monthly mean over land than over water. Both of

these effects could suppress or enhance any real land-ocean contrast in the AOD. As well as continued refinement of AOD retrieval algorithms, it is recommended that future work attempts to quantify the potential magnitude of these sampling effects on land/ocean contrast, which have received comparatively little attention to date. Finally, the satellite data also indicate a larger spatial extent to the aerosol loading over the ocean compared to ALADIN-Climate. This difference could

be due to possible overactive aerosol deposition in our simulations. A specific section (4.2.2) is dedicated to using ACAOD products for evaluating the model over the ocean.

In order to make more robust comparisons over the continent, a second box (box_S, 15-25°E / 5-15°S, see Figure 1) is defined over biomass burning sources. Figure 4a indicates the daily-mean AOD at 550 nm averaged over box_S from MODIS, MERRA-2 and ALADIN-Climate. There is a

good agreement, with monthly means of 0.59, 0.62 and 0.49 for MODIS, ALADIN-Climate and MERRA2, respectively. The model is very consistent with the MODIS retrievals and slightly higher





than MERRA2. Differences between MERRA2 and ALADIN-Climate may reflect the different model biomass burning emissions. Compared to MODIS, a small mean bias of +0.04 is found in the model over smoke sources. In terms of temporal correlation, the score is worse compared to

MODIS. This could be due to the time frequency of biomass burning emission imposed in the ALADIN-Climate model. The absence of spectral nudging in the ALADIN-Climate simulation can also explain part of the low temporal correlation, which is clearly higher (0.73) for MERRA-2 data. Data from three AERONET stations (Mongu (Zambia), Lubango (Angola) and Ascension Island) provide additional AOD evaluation. Figure 5 indicates the daily-mean AERONET and ALADIN-

Climate (only daytime values) AOD at each station. At Lubango, the model is able to correctly simulate the AOD, except in the beginning of September when it is overestimated. The maxima (AOD of 0.6) of 20 and 27 September are also well represented by the model. The total monthly-mean AOD simulated by ALADIN-Climate (0.41) is consistent with AERONET (0.43), if with a small mean negative bias. Over the Mongu station, the comparisons indicate a more pronounced

negative bias (mean value of -0.14). This is due to a nearly constant underestimate of total AOD throughout the period of the simulation, leading to a simulated monthly-mean of 0.47, lower than the one observed (0.61). In parallel, the two maxima detected in AERONET data are well captured by the model. The first one, occurring between the 18 and the 21 September, is simulated too early and its magnitude is overestimated by about ~0.2. The second (23-29 September) is better

reproduced in terms of magnitude (~ 1.0) but a significant underestimate in its duration is observed. Indeed, the BBA event starts around the 22 September in the observations, while it is simulated between 26 and 29 September by the model. Finally, the model is able to correctly simulate the magnitude of AOD along the transport over SAO to the remote location of Ascension Island (Figure 5). The simulated monthly-mean value (0.26) is comparable to AERONET (0.21) with a small

positive bias, primarily because of overestimates during the 06-10 September period. This suggests that winds and aerosol deposition are also well represented. To summarize, the analysis of AOD comparisons demonstrate that the ALADIN-Climate model reasonably simulates the magnitude of AOD during September 2016, even if some biases are detected, especially over the ocean (a negative bias, primarily near the coast) when compared to satellite data.

**4.2.2 ACAOD over SAO**

Due to the significant presence of Sc over SAO, the use of satellite clear-sky AOD products as a model evaluation tool is limited. This limitation is overcome with new retrievals of ACAOD from MODIS and OMI, summarized in Table 2. Figure 6 indicates consistent estimates of monthly-mean ACAOD between ALADIN-Climate, MODIS-DB, MOD06ACAERO and OMI. For all

independent estimates, Figure 6 indicates values of about ~0.4-0.5 (550 nm) near the Angola coast.





ACAOD then decreases to ~0.2 over SAO. ACAOD is underestimated by ALADIN-Climate over the ocean, especially when compared to the two different MODIS products. Indeed, MODIS-DB and MOD06ACAERO data reveal a larger regional extent over the ocean compared to the model. The ACAOD extent is less pronounced in the OMI data, which is thereby more consistent with
ALADIN-Climate.

Additional comparisons were performed over box_O using MODIS-DB products. Figure 4b indicates daily-mean model ACOAD obtained for the SMK and SMK_SN (not discussed in this part) simulations and MODIS-DB. The monthly-mean ACAOD is underestimated by the model (SMK configuration, red dotted line) with a mean value of 0.20 (at 550 nm), 0.06 less than MODIS-
DB (0.26). The ALADIN-Climate simulation underestimates two maxima observed by MODIS-DB, around 03-05 and 20-24 of September, explaining part of the negative bias (-0.06) in the SMK simulation. For the rest of the time period, we observe a realistic estimation of ACAOD by the model. It should be mentioned that additional analyses of the simulated ACAOD is also discussed in Shinozuka et al. (2018) for different boxes defined in the ORACLES-1 program. As for AOD,
ALADIN-Climate is shown here to simulate realistically the concentration of aerosols transported above clouds over SAO. This allows to address the SW DRF exerted by BBA at TOA during ORACLES, after investigating the vertical structure (part 4.2.3) and SW absorbing properties (part 4.2.4).

**4.2.3 BBA Vertical Structure**

4.2.3.1 ALADIN-Climate Extinction vertical profiles

The vertical distribution of the modeled BBA extinction is analyzed over the continent and SAO in Figure 7 as the monthly-mean extinction vertical profiles (at 550 nm) for two different transects at latitudes of 8 and 15°S. For both transects, the highest extinctions are identified over the continent, close to biomass burning sources, with extinctions ~0.2 km⁻¹. The amplitude of BBA extinction
coefficient (top and bottom right panels) decreases during the transport, reaching values of ~0.05-0.10 km⁻¹ (at 8 and 15°S, respectively) for longitudes near 0°. For both profiles, the top of the smoke plume is around ~5000 m over the continent in the simulations, very consistent with the altitude of the top plume over the continent reported by Das et al. (2017) from Caliop lidar observations.

Over the SAO, two different well-distinguished aerosol layers are simulated ; a first one mainly located in the MBL and mostly due to primary sea spray aerosols and a second BBA layer located above, between 2000 and 4000 m. The top of the marine aerosol plume is simulated around ~1000 m in the model and is separated from the smoky layer by a clean atmospheric layer, especially at 15°S, characterized by extinction near ~0.05 km⁻¹. For both transects, the top of the smoke plume



decreases from 10°E to 10°W, starting around ~5000 m near the coast to reach ~4000 m at 10°W. This BBA stratification over SAO is consistent with the vertical structure reported by Das et al. (2017) for latitudes comprised between 0 and 10°S. Indeed, they report a transport of smoke that mainly occurs between 2000 and 4000 m over SAO, contrary to the different models used in this study, which both indicate a more pronounced decline of the altitude of BBA during the transport.

This result is also consistent with a previous study from Haywood et al. (2003) over this region.

The elevated plume is mainly composed by BBA characterized by a decrease in extinction during the transport from 0.15 km$^{-1}$ (at 15°E) to 0.08 km$^{-1}$ (near 0°) for the transect at 8°S. Such extinction values are consistent with those reported by Das et al. (2017), who indicated Caliop extinction around ~0.1-0.15 km$^{-1}$ over SAO (for latitudes between 0 and 10°S). The extinction due to BBA is

negligible in the simulation for longitudes highest than ~10°W, especially for the transect at 15°S. For both transects reported in the Figure 7 (left panels), significant extinctions are simulated within the MBL, with values of about ~0.2-0.25 km$^{-1}$ mostly due to the presence of primary sea-spray aerosols. Figure 7 indicates the highest values at 8°S compared to 15°S.

Based on the transects, no favorable conditions are identified allowing an efficient mixing of BBA

within the MBL during the transport of aerosols over SAO. Such results are contradictory to the schematic view of Gordon et al. (2018), who proposed that an efficient mixing of smoke only occurs around 0° E - 10° W within the MBL. Advections processes are absent in our simulations and could limit the possible impact of BBA on cloud droplet concentrations and Sc properties. In the ALADIN-Climate simulations, smoke aerosols primarily remain above the MBL during

transport, with little vertical mixing. For this reason, in the model, the impact of BBA on the Sc microphysical/optical properties will be primarily through the semi-direct radiative effect.

4.2.3.2 Comparison with HSRL-2 Extinction

Aerosol extinction coefficients derived at three different wavelengths (355, 532 and 1064 nm) by the HSRL-2 instrument permit local 1-D comparisons with ALADIN-Climate simulations. In

addition to evaluating the simulated extinction vertical profiles, the spectral dependence of the model calculated extinction can also be evaluated. Figure 8 reports the extinction vertical profiles for three different days (12, 22 and 24 September), as well as RH profiles obtained from MERRA2 and ALADIN-Climate (black dotted and solid lines, respectively). Those specific days have been chosen to represent different locations within Box_O (Figure 1). The vertical profile of CF

simulated by ALADIN-Climate (yellow dotted line) is also included in Figure 8. It should be mentioned that the wavelengths of ALADIN-Climate are not exactly the same as those from HSRL-2, especially for the UV spectral band (355 and 440 nm, respectively). In addition and due to the





significant CF, HSRL-2 data are not necessarily available near the surface and remain above cloud top (~2000 m) in most cases.

For September 12, our simulations indicate that the vertical structure of the BBA plume (dashed red, purple and blue lines) is not well represented in the model even though both the model and HSRL-2 place most of aerosol above the MBL. At both times (11:00 and 13:00 UTC), the aerosol extinction coefficients simulated by ALADIN-Climate are overestimated (underestimated) for altitudes between 1500-3000 m (3000-6000 m). This compensation of errors leads to a consistent

averaged (between 1500 and 6000 m) integrated extinction in the simulation (0.07 km$^{-1}$ at 550 nm) compared to HSRL-2 observations (0.06 km$^{-1}$) at 13:00 UTC but significant underestimates at 11:00 UTC (0.05 and 0.13 km$^{-1}$, for ALADIN-Climate and HSRL-2, respectively). In addition, we observe important biases in the simulated RH, especially at 11:00 UTC between the surface and 3000 m (positive bias) and (negative bias) above 3000 m.

For the 22 and 24 September, Figure 8 indicates that the altitude of the smoke plume over SAO is realistically represented by ALADIN-Climate, especially for the plume located between 3000 and 6000 m. However, a second aerosol layer, which is observed from HSRL-2 between 2000 and 3000 m for September 24 at 11:00 UTC (Figure 8e) is absent in ALADIN-Climate and simulated below (between 1000 and 2000 m). For the same day at 12:00 UTC (Figure 8f), similar conclusions are

obtained with a maxima at ~3500 m which is well simulated by ALADIN-Climate but the second plume (observed around ~2000-2500 m from HSRL-2) is totally absent in the model. For both days, Figure 8 reveals that the magnitude of the simulated extinction is generally underestimated compared to HSRL-2, true at each wavelength. As an example, for 24/09 at 11:00 UTC (Figure 8a), the local maxima (~0.30 km$^{-1}$) derived by the HSRL-2 instrument at 5000 m is significantly lower

(~0.15 km$^{-1}$) in the model. This is also observed for the second aerosol plume at ~2500 m for that day. Such conclusions can be drawn for all cases (12, 24 and 26 of September), with a negative (mean) bias (indicated only at 550 nm and for the whole atmospheric column) of between -0.01 and -0.08 km$^{-1}$. This could be attributed to incorrect smoke emissions, e-folding time, OC to POM ratio, optical properties (especially the mass extinction efficiencies) of BBA, as well as the different

parameterizations used for representing hygroscopic properties of aged smoke. In section 4.3, specific attention is paid to the impact of RH transported within the smoke plume on BBA extinctions.

**4.2.4 BBA (SW) Absorbing properties and Heating rate**

4.2.4.1 Absorbing properties at the biomass burning source

The magnitude and the sign of the DRF of BBA exerted over SAO is highly sensitive to the smoke SSA. The monthly-mean (whole-column integrated) SSA (for the fine aerosols) simulated by



ALADIN-Climate for September 2016 (Figure S3 in Supplement) indicates values of about ~0.85 (at 550 nm) over a large part of the subcontinent. SSA increases near the coast (~0.89-0.90) and during transport over the SAO (~0.92 to 0.95). Local comparisons at the two continental

AERONET stations (Mongu and Lubango, Figure S3 in Supplement) reveal a good agreement between the simulated and observed SSA, characterized by low bias of about +0.01/-0.02. A larger negative bias is observed and documented at the Lubango site. However, the day to day variability is not represented in the model and SSA is nearly constant (~0.83-0.84) in the simulation. As an example, the lowest values (~0.79-0.80) detected by AERONET are absent in the model. The same

conclusion is obtained for the highest values derived from observations, especially at Lubango (Figure S3 in Supplement). However, such results indicate the ability of the model at reproducing absorbing properties of BBA close to biomass burning emissions with limited bias (-0.02/+0.01).

4.2.4.2 Absorbing properties over SAO

The model comparison to in-situ surface-based SSA values at Ascension Island reveals more

discrepancy. Figure 9 shows the daily-mean SSA obtained at the surface from in-situ observations and calculated with ALADIN-Climate at two different altitudes (0.2 and 3 km). The model is not able to reproduce the low values (mean of 0.87) observed at the surface. Indeed, near the surface, the simulation indicates a simulated SSA of nearly 1 for all of the September 2016 period. The model MBL optical properties are mainly controlled by primary marine aerosols (see Figure 7)

leading to SSA close to unity. This highlights also that the mixing of BBA within the MBL is possibly underestimated in the model, although LASIC observations also show little smoke is present at the surface in September (Zuidema et al., 2018).

It should be noted that the low values of SSA obtained at Ascension island reflect long-term ageing processes for the BBA that are not currently included in ALADIN-Climate. This chemical process

could increase the absorbing efficiencies of BBA (Fierce et al., 2016) during the transport (decrease of SSA) and could explain the opposite results obtained in the model, which simulates an increase of SSA (not shown, Figure S3) from biomass burning sources to SAO.

4.2.4.3 SW Heating Rate

Figure 10 indicates the SW heating rates only due to BBA for two transects defined at latitudes of 8

and 15°S, similar to Fig. 8. The effect of BBA is isolated by subtracting the heating rates in the simulations without BBA from those with BBA. Significant additional SW heating is simulated over the continent and between 2 and 4 km over SAO due to the presence of absorbing smoke. Over the continent, the additional heating is about ~1 °K by day with maxima near ~1.5 °K by day for altitude of ~4000m at 8°S. The simulated heating is approximately 1°K/day near the coast,

decreasing to 0.5°K/day during transport. For both transects, SW heating occurs mainly between 2



and 4 km over SAO. At 15°S, the SW heating is less pronounced than at 8°S, in agreement with the difference in the extinction profiles (see Fig. 7). The SW heating due to BBA absorption is clearly visible only above the MBL and there is no clear additionnal SW heating within it.

Such values of SW heating due to smoke appear to fall well within the range of values reported by

Tummon et al. (2010), Gordon et al. (2018), Adebiyi et al., (2015) or Wilcox et al. (2010), who reported, respectively, additional SW heating due to smoke of 1 (JJAS period), 0.34 (5 days of simulations), 1.2 (for fine AOD > 0.2) and 1.5 °K by day. In addition, Keil and Haywood (2003) estimated a SW heating rate of 1.8°K/day near the coast using a radiative transfer model and observations during SAFARI-2000. The temperature change (estimated through two parallel

simulations including or not smoke) due to BBA is about +0.5-0.8 °K between 2 and 4 km (Figure S4 in Supplement material) in a good agreement with the value (+0.5 °K) of Sakaeda et al. (2011) or more recently proposed by Gordon et al. (2018).

ORACLES SW heating rates retrieved from the SSFR retrievals of SSA and ASY (see Part 3.2.3.2) in conjunction with HSRL2 extinction profiles (Part 3.2.3.1) are also used to assess our simulations.

Figure 11 indicates the instantaneous (12:00 UTC) SW heating (only due to aerosols) vertical profiles obtained for 20 September from SSFR and ALADIN-Climate. Two ALADIN-Climate heating profiles indicate clear-sky and all-sky conditions. Figure 11 indicates that the location of the additional SW heating due to BBA is well represented by the model, with a notable increase between 3 and 4 km, in agreement with the SSFR retrievals. SW Heating between 2 and 2.5 °K/day

are simulated at these altitudes with the highest values obtained under all-sky conditions (black dashed lines). However, significant underestimates are observed within the smoke layer, where SSFR observations indicate SW heating of about ~3 to 3.5 °K/ day. As the Sc COD is found to be consistent between simulations and the SSFR cloud retrievals (COD ~9), we hypothesize that the difference in SW heating is due to local underestimates of aerosol extinction as well as of BBA

absorption in the model. A second aspect concerns the large underestimate of SW heating around ~1.5 km in the ALADIN-Climate simulation. Indeed, the local maxima of ~2.5 °K/ day obtained from SSFR observations is totally absent in the model, but can be traced back to another layer detected by HSRL.

**4.3 Impact of the RH transported within the smoke plume on optical properties**

As mentioned previously, a specific simulation (SMK_SN) that includes the method of spectral nudging (Radu et al. 2008) was also performed. The nudging is applied to wind vorticity and divergence, surface pressure, temperature and specific humidity, using a constant rate above 700 hPa, a relaxation zone between 700 and 850 hPa, while the levels below 850 hPa are free. This simulation was motivated by different studies (Haywood et al., 2003, Adebiyi et al., 2015) that



indicated a correlation between BBA and specific humidity. In these studies, biomass burning plumes are associated with specific humidities greater than 2 g.kg$^{-1}$, while outside the smoke plumes the values are less than 1 g.kg$^{-1}$. To date, few regional climate modeling studies have investigated the humidity's potential role on the optical properties of smoke, that could, in turn, impact the DRF exerted by BBA. This ALADIN-Climate simulation (SMK_SN) addresses this

specific point.

Figure 4b showing the daily-mean ACAOD (averaged over the box_O) from MODIS-DB along with the simulations through comparing the ACAOD with and without the nudging of RH, indicates an impact. For SMK_SN, the negative bias is reduced compared to MODIS-DB and equals to -0.02 (biais of -0.06 for the SMK run). The maxima in ACAOD observed between 19 and 25 September

is better reproduced in the SMK_SN simulation, consistent with a slightly improved temporal correlation (0.42).

In addition to the satellite observations, we also used HSRL-2 vertical profiles of extinction (already presented in Figure 8) to investigate the impact of RH on BBA optical properties. Figure 12 shows, for 24 September only, the vertical profiles of RH by MERRA-2 and ALADIN-Climate

(SMK and SMK_SN simulations), as well as aerosol extinction (at 550 nm) from HSRL-2 and the model (SMK and SMK_SN). A significant improvement is evident in the SMK_SN RH vertical profiles, reducing the bias with MERRA2, especially at 09:00 and 12:00 UTC. For each case, Figure 12 indicates that RH is better represented in SMK_SN especially at altitudes where the transport of smoke occurs, i.e., between 2000 and 5000 m (Figure 7).

These changes in RH profiles in SMK_SN run impact the BBA optical properties for the different cases, notably by increasing extinction within the smoke plume and a remarkable agreement in extinction is observed between HSRL-2 and the SMK_SN simulation. As an example, at 09:00 UTC, the improvement of the simulated RH between 3500 and 6000 m in SMK_SN significantly reduces the bias in the simulated extinction at those altitudes. At 4500 m, the simulated extinction is

very consistent with HSRL-2, with maxima around ~0.2 km$^{-1}$. Similar conclusions are also observed for the other cases presented in Figure 12. At 11:00 UTC, important improvements are found for altitudes between 2000 and 5000 m in the SMK_SN simulation compared to SMK. At this time, a negative bias persists between 2000 and 3500 m, marked by a bias in RH even in the nudged simulation. These results are consistent with the study of Adebiyi et al. (2015) who used CALIPSO

smoke extinction profiles to show that the largest extinction coefficients co-occur with high RH (~80%) at the top of the BBA layer. As discussed recently in Kar et al. (2018), this increase could be due to enhancement of the size of aged smoke during the transport over SAO.



A second important aspect of these results concerns the possible overestimates of the increase of extinction with RH as parameterized in the present version of ALADIN-Climate. As indicated for both cases, excellent agreement is generally observed in the extinction profiles even if some slight negative bias in RH remain. This can be clearly detected at 4500 m (09:00 UTC) or 4000 m (11:00 UTC). At 12:00 UTC and for altitudes between 1000 and 2000 m, RH simulated in SMK_SN is consistent with MERRA2 data, while the simulated extinction is overestimated.

These original results, using for the first time coincident in-situ observations and nudged simulations (allowing the capture of the elevated humidity transported within smoke plume) of aerosol extinction within BBA plume, clearly indicate the significant impact of RH on BBA optical properties. This underlines the importance of including in models the fire processes related to the presence of humidity in the smoke plume over SAO. A second important aspect concerns the presence of possible errors in the actual parameterization used in ALADIN-Climate to calculate the evolution of BBA extinction with RH. In that sense, nudged simulations, associated with in-situ data obtained during ORACLES would certainly provide a unique opportunity to test and constrain the hydrophylic properties of BBA over SAO. Future work will extend significantly the number of cases studied to test the robustness of these first results.

**5. Direct Radiative Forcing and impact of BBA**

**5.1. DRF exerted at TOA**

The monthly-mean DRF exerted at TOA (in the visible spectral range) is indicated in the Figure 13 in clear-sky (left) and all-sky (right) conditions. The ALADIN-Climate estimates do not include possible SST adjustments due to BBA radiative effects, even if this could be important over SAO (Sakaeda et al., 2011). Figure 13 indicates an important regional gradient in the sign of DRF over the domain in all-sky conditions, with a rather negative forcing over the continent (net cooling) and positive (net heating) over SAO. Over the continent, the mean DRF is found to be mostly negative ($\sim$-5/-15 W.m$^{-2}$) over Angola, with local maxima up to -20 W.m$^{-2}$. An interesting result concerns the presence of significant positive forcings along the coast from Gabon to Namibia, with values of $\sim$+10 to +20 W.m$^{-2}$. Such significant positive forcing at TOA are correlated with both the presence of Sc along the coast of Angola, Republic of Congo and Gabon (see Figure S5 in Supplement material) and the high surface albedo over Namibia (Figure 1).

On the contrary, over SAO, DRF exerted at TOA is found to be mainly positive in all-sky conditions in agreement with a large literature focused on this region (Meyer et al., 2013; Feng and Christopher, 2015; De Graaf et al., 2012, 2014; Zuidema et al., 2016). The impact of the presence of Sc on the sign of DRF at TOA is clearly shown when comparing the ALADIN-Climate simulations in clear-sky and all-sky conditions. The large cooling effect at TOA is replaced by a significant





heating over a large part of SAO. However and when averaged over the same region (20°S–10°N and 10°W–20°E) as defined in Feng and Christopher (2015), an important underestimate is detected compared to satellite observations. Indeed, the instantaneous (at satellite time overpassing)
monthly-averaged DRF is found to be about +6 W.m$^{-2}$ in ALADIN-Climate and ~+35 W.m$^{-2}$ in the study of Feng and Christopher (2015). A better agreement is obtained with Oikawa et al. (2013), who reported an annual mean of +3 W.m$^{-2}$ over Southern Africa using CALIPSO and GCM simulation. More recently, Gordon et al. (2018) indicate a regional DRF of +11 W.m$^{-2}$ at TOA close to the one obtained in this study, but for five smoky days. We suspect the underestimate of LCF
(Figure 2b) to be mostly responsible for this large difference with the Feng and Christopher (2015) estimates. Finally, comparisons with the climatological estimates based on MACC NRT data for the period 2010-2015 (Figure S6 of Supplement material) indicate important differences. Figure S6 indicates that the positive DRF simulated by ALADIN-Climate is absent in the MACC NRT data, except locally over the continent.

**5.2. Impact on the continental surface energy budget and dynamics**

The potential impact of BBA on the « continental climate » has been investigated by using the differences between the CTL and SMK simulations. Figure 14 shows the monthly-mean difference (September 2016) of the following variables; surface net SW radiations (upper left), 2-meter temperature (T2m, upper right), sensible heat fluxes (SHF, bottom left) and the PBL height (bottom
right). The potential effect of BBA on the continental precipitation is not studied as little or no precipitation occurs south of approximately 8°S during the austral winter season.

Smoke aerosols are responsible of an important dimming of about -30 to -50 W.m$^{-2}$ (monthly mean) over the continent and -10 to -40 W.m$^{-2}$ over SAO during September 2016, with the highest impact logically located over smoke sources. Such estimates are consistent with those reported by Sakaeda
et al. (2011) or Tummon et al. (2010). This impact of BBA results in an important decrease in the T2m over Congo, Angola and Zambia, as well as certain regions of Southern Africa. The impact is approximately ~-1 to -3 °C over the continent, in good agreement with the values reported by Sakaeda et al. (2011). When averaged over box_S (5-15° S / 15-25° E), the impact of BBA on T2m is about -1.7°C during September 2016 (Figure 15a). The daily-mean impact of BBA remains
constant during this period, except for the end of September when the effect is negligible. For the 26 to 31 September, we hypothesize that compensations should occur, the « dynamical » effect of BBA being more important than the « dimming » effect. As mentioned previously and contrary to Sakaeda et al. (2011), the impact of BBA on SST is not quantified as the ALADIN-Climate simulations have been performed with prescribed SST.



In parallel with the changes in surface temperature, the sensible heat fluxes (SHF, Figure 14, bottom
left) significantly decrease, meaning weaker fluxes, over almost the entire subcontinent, with
maxima in the main biomass burning sources. The decrease is about -20 to -30 W.m$^{-2}$ over the
continent, with a mean value of -25 W.m$^{-2}$ when averaged over the box_S (Figure 15b). The impact
of smoke is important throughout the whole time period, with SHF changing from a (monthly-

mean) value of 85 W.m$^{-2}$ to 60 W.m$^{-2}$ for the CTL and SMK runs, respectively. This is consistent
with the findings of Sakaeda et al. (2011) or Tummon et al. (2010). Indeed, the latter report a
decrease over almost the entire subcontinents, with a maximum (decrease of ~50%) in the main
smoke region. As a result of the significant decrease in T2m and SHF over much of the
subcontinent, the PBL height also decreases in the SMK run. This decrease is significant over much

of the subcontinent, in accordance with the results of Tummon et al. (2010), with regional maxima
up to ~ -400 m (Figure 14, bottom right). The lowest changes are observed along the west coast
between 10 and 15°S and small regions of increased PBL height occur over southern Namibia and
Northern Angola consistent with increases in the surface temperature. When averaged over the
continent (box_S), the decrease in the PBL height is also significant, changing from ~1200 to ~1000

m (monthly-averaged) for the CTL and SMK simulations (Figure 15c), respectively. The decrease is
nearly constant during September 2016, except for 25 to 30 September (decrease), and not
necessarily correlated with AOD. The difference in the PBL height can reach a maximum decrease
of 300 m (Figure 15c).

**6. Conclusion**

The transport, vertical structure, SW radiative heating, SW direct radiative forcing and climatic
impact exerted by absorbing BBA in the SAO have been estimated for September 2016 using the
ALADIN-Climate model in the context of the ORACLES and LASIC projects. The model is able to
represent LWP and COD well, although with a large underestimate in LCF. The simulated BBA
AOD is consistent over the continent (~0.7 at 550 nm) compared to MERRA2 or MODIS data and

also locally against AERONET data. We have also used new recent retrievals of ACAOD (OMI or
MODIS) to demonstrate the ability of the model to reproduce reasonable values of smoke
concentrations above Sc during the transport over SAO.

The simulations indicate the transport of BBA over SAO mainly occurs between 2 and 4 km,
consistent with aircraft lidar observations. There is some indication that the entrainment of BBA in

the MBL could be underestimated by the model contrary to the recent litterature (Zuidema et al.,
2018). This possible bias could lead to underestimate the BBA indirect forcing in this ALADIN-
Climate configuration. In parallel, the absorbing properties (SSA) of BBA are consistent over
biomass-burning sources compared to AERONET but significantly higher when compared to



Ascension Island (LASIC) surface observations. The significant difference could be due to the absence of internal mixing treatment in the model, a lack of representation of the long-range aging processes, and/or the absence of mixing of BBA in the MBL. In addition, the important SW absorption by BBA produces an additional SW heating of ~1 °K/day.

The ALADIN-Climate simulations reveal a significant regional gradient in the sign of the SW DRF at TOA (all-sky conditions and fixed SST conditions), with mostly negative (continent) and positive (SAO) forcing, mainly due to changes in the underlying albedo associated with highly absorbing BBA. Over the continent, an intense monthly-mean positive forcing ($+10/+15$ W.m$^{-2}$) is simulated over the Gabon, part of the Congo and Angola, mainly due to the presence of low Sc. Over SAO, a DRF of $+6$ W.m$^{-2}$ (20°S–10°N and 10°W–20°E) is simulated at TOA during all the ORACLES-1 period.

One of the main original results concerns the use of coincident in-situ observations and nudged simulations (allowing to capture the elevated humidity transported within smoke plume) of aerosol extinction within BBA plume. Results highlight the significant effect of enhanced moisture on BBA extinction that considerably reduces the negative bias (in the simulated extinction) in the nudged simulation (SMK_SN) with ORACLES-1 data, compared to the SMK (no nudging) run. A second important aspect concerns the possible errors in the actual parameterization used to estimate the changes in BBA extinction with RH in the model. Indeed, our results indicate a possible overestimate of the increase in smoke extinction due to RH when compared to ORACLES-1 observations. Nudged simulations, associated to in-situ observations, would certainly provide a unique data-set to test and constrain the hygroscopic properties of BBA over SAO. All of these points have possible implications for DRF considerations and future works will extend significantly the number of cases studied to test the robustness of the results.

For September 2016, the important negative surface dimming due to BBA (around -5 to -15 W.m$^{-2}$) over the subcontinent significantly modifies the surface energy budget over much of Southern Africa. Indeed, the decrease in the net surface SW radiations is compensated by a decrease in sensible heat fluxes (-25 W.m$^{-2}$, monthly mean) and surface land temperature (-1.5 °C) over Angola, Zambia and Congo notably. The association of the surface cooling and the lower tropospheric heating tends to decrease the continental PBL height over the continent by about ~200 m.

Finally, the indirect radiative effect exerted by BBA remains to be investigated with the ALADIN-Climate model using ORACLES/CLARIFY/AEROCLO-sA data in a manner similar to that presented here for DRF considerations. Once evaluated for all forcings over this region, long-term simulations (ERA-INT period) will be done to assess the possible feedbacks of BBA on Sc properties and the regional radiative budget at a climatic scale.



**Author contribution**

PN and MM designed the ALADIN-Climat configuration and MM performed the simulations. RR, DSM, PN developed the ALADIN-Climat model. AMS, MS, KM, HJ, OT and CH have participated to provide the different aerosol satellite products. PZ provided in-situ LASIC (Ascension Island) observations; SS and SC data from the SSFR instrument. SB and RF participated to provide the HSRL-2 data. JR, RW, PS, PF and all co-authors have participated to the analysis of all observations and simulations. MM finalised the manuscript with contributions from all co-authors.









**Acknowledgment**

This work has been conducted in the frame of the ORACLES investigation under the National Aeronautics and Space Administration's Earth Venture program. The LASIC extinction and SSA values are available through doi:10.5439/1369240, obtained from the Atmospheric Radiation Measurement (ARM) User Facility, a U.S. Department of Energy (DOE) Office of Science user

facility managed by the Office of Biological and Environmental Research. PZ acknowledges support from DOE ASR grant DE-SC0018272. AERONET data are available from https://aeronet.gsfc.nasa.gov. MODIS data are available from the NASA Level 1 and Atmospheres Data Center at https://ladsweb.modaps.eosdis.nasa.gov/. MODIS above-cloud data products are available from the authors on request. MISR data are available from the NASA Atmospheric

Sciences Data Center at https://eosweb.larc.nasa.gov/. M. Mallet, P. Nabat and P. Formenti acknowledge the AErosols, Radiation and CLOuds in southern Africa (Aeroclo-sA) project, supported by the French National Research Agency under grant agreement n° ANR-15-CE01-0014-01, the French national programme LEFE/INSU, the Programme national de Télédetection Spatiale (PNTS, http://www.insu.cnrs.fr/pnts), grants n° PNTS-2016-02 and PNTS-2016-14, the French

National Agency for Space Studies (CNES), the Centre National de la Recherche Scientifique (CNRS), the South African National Research Foundation (NRF) under grant UID 105958, and the European Union's 7th Framework Programme (FP7/2014-2018) under EUFAR2 contract n°312609.






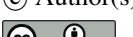



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



| Aerosol Species | $r_0$ | $\sigma$ | Density | n / k | MEE | SSA | ASY |
|---|---|---|---|---|---|---|---|
| *Fresh smoke* | *0.10* | *1.30* | *1.350* | *1.50 / 0.03* | *4.05* | *0.84* | *0.51* |
| *Aged smoke* | *0.12* | *1.30* | *1.350* | *1.50 / 0.03* | *5.05* | *0.90* | *0.58* |

*Note. Here $r_0$ and $\sigma$ are the median radius (in µm) and geometric standard deviation of the lognormal distribution. Mass density is reported in g.cm$^{-3}$, m is the complex refractive index, and MEE and SSA the mass extinction efficiency and single scattering albedo in dry state and reported at 550 nm.*

**Table 1.** Parameters describing aerosol components used in the ALADIN-Climate model for the two smoke tracers and the resulting optical properties.




|  | Aerosols | Clouds | Spatial / Temporal Resolutions |
|---|---|---|---|
| MOD06ACAERO | ACAOD | # | 0.1× 0.1 ° / daily |
| MODIS DB | ACAOD | # | 0.5 × 0.5 °/daily |
| MODIS DT/DB | AOD | LWP, COD | 1 × 1 ° / daily |
| OMI | ACAOD | # | 0.5 × 0.5 ° / daily |
| MISR | AOD | # | 0.5 × 0.5 ° / monthly |
| SEVIRI | # | LWP, COD | 0.5 × 0.5° / daily |
| ERAI | # | LWP, LCF | 0.7 × 0.7 ° / daily |
| MERRA2 | AOD by species | # | 0.5 x 0.625 ° / hourly |
| MACC | DRF TOA (all-sky) | # | 1.125 × 1.125 ° / daily |

**Table 2.** Satellite and reanalysis data used in this study to analyse aerosols and Sc clouds microphysical/optical properties.











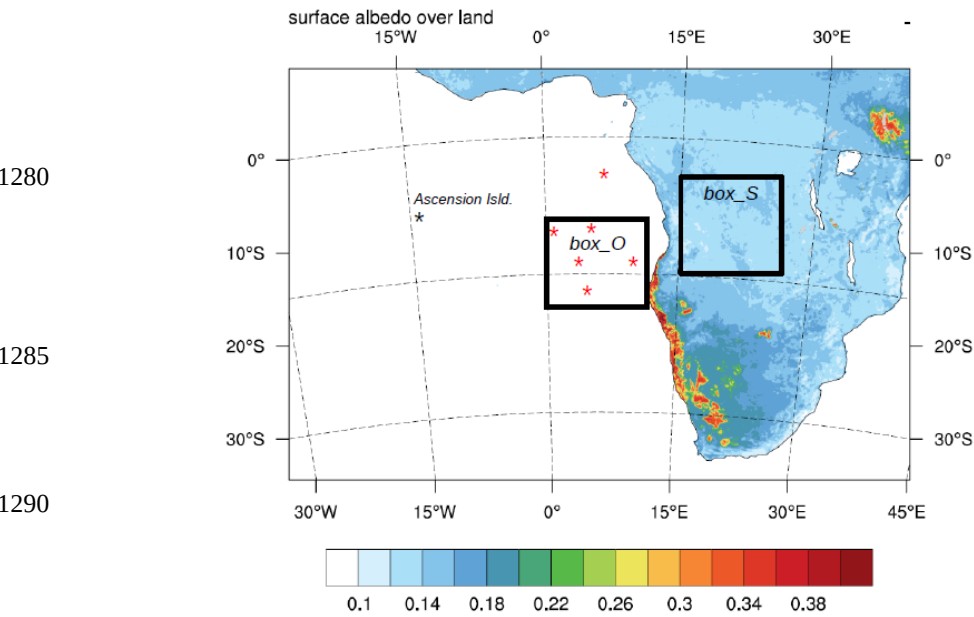


**Figure 1.** Domain defined for the ALADIN-Climate model simulations (here, the surface albedo is represented). The two different boxes (box_O and box_S) and the Ascension Island are indicated. Red stars represent the localisation of different profiles studied in the Part 4.2.3.2.















**Figure 2.** Daily-mean Sc properties (LWP, LCF and COD) simulated by ALADIN-Climate and from ERA-INT and SEVERI data. Values have been averaged over the box_O.





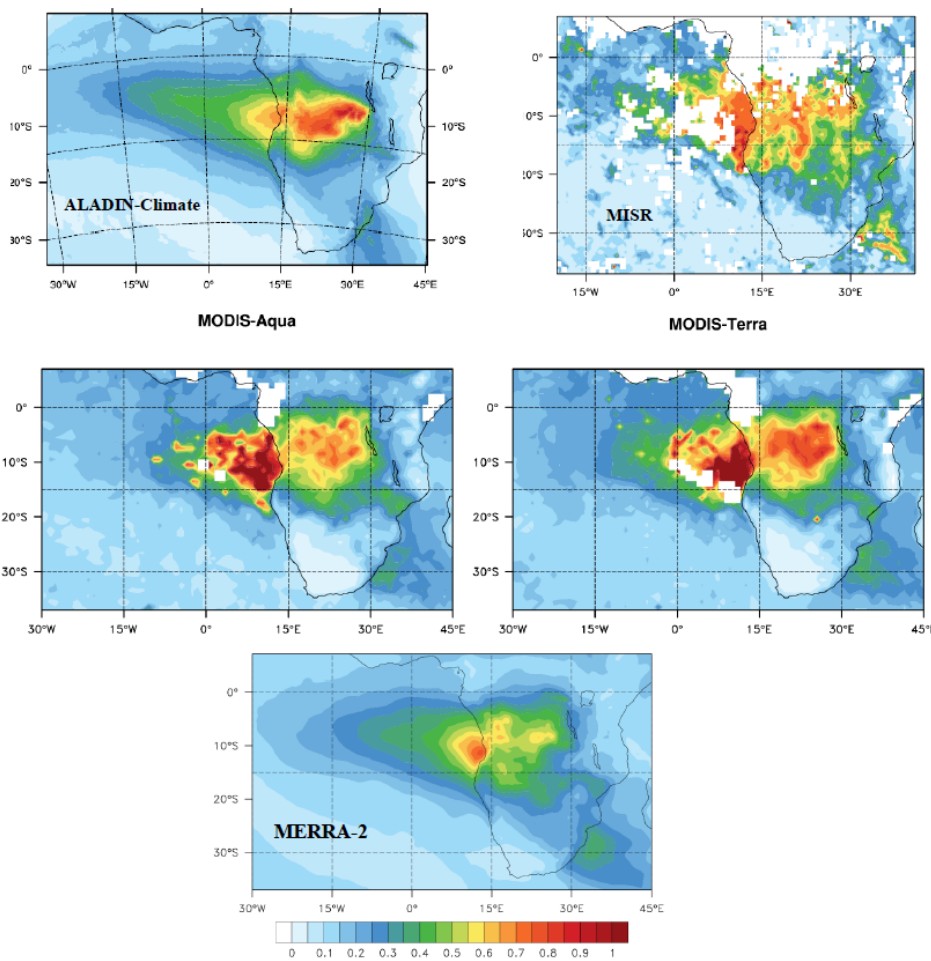

**Figure 3.** Total monthly-averaged Total AOD (at 550 nm) for september 2016 simulated by ALADIN-Climate, MERRA2 and derived from the MODIS Terra, Aqua (combined DB/DT products) and MISR sensors.






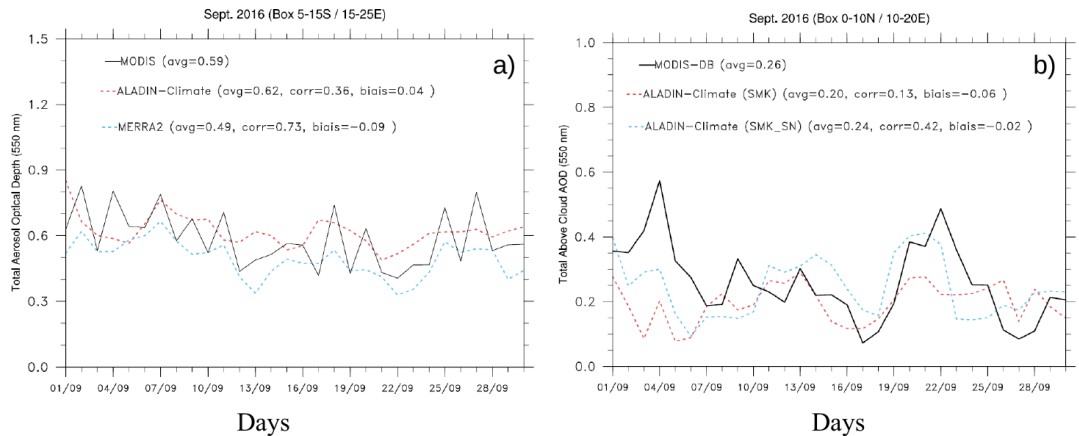

**Figure 4.** Total daily-mean AOD (at 550 nm) averaged over the box_S (5-15S / 15-25E) from the ALADIN-Climate model, MERRA2 and derived from MODIS-Aqua DT data (left). Daily-mean ACAOD (at 550 nm) estimated from the MODIS-DB satellite and two different configurations of the ALADIN-Climate model (SMK and SMK_SN) averaged over the box_O (0-10N / 10-20E) 1365 (right).






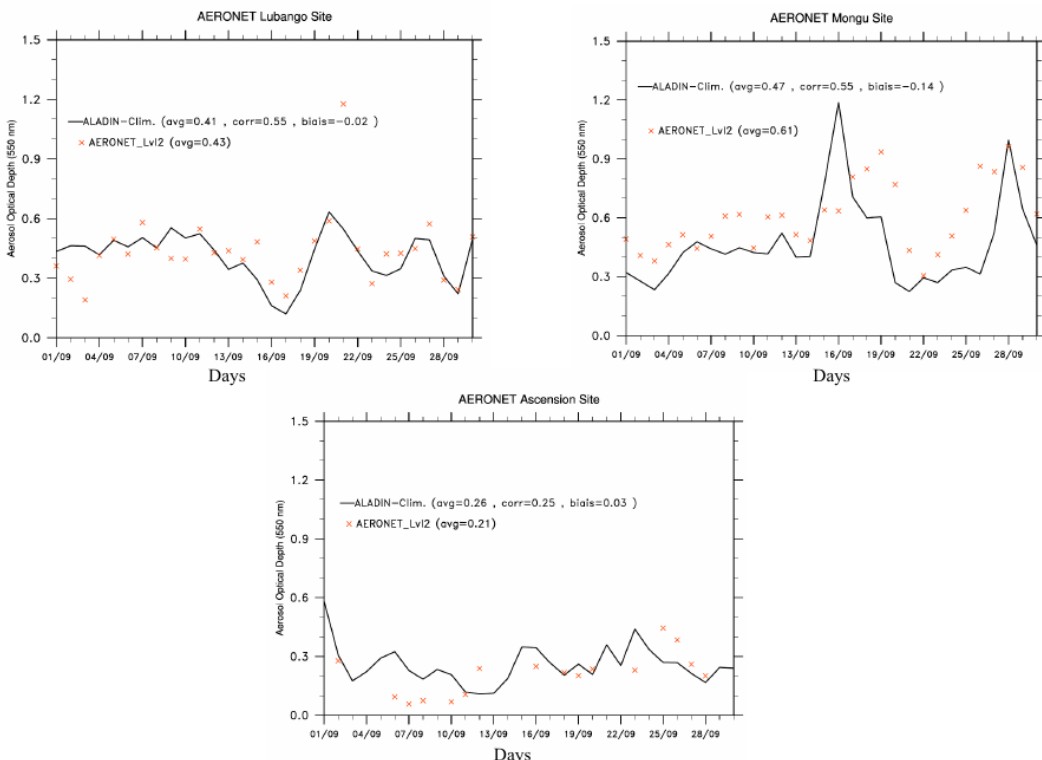

**Figure 5.** Comparisons of daily-mean total AOD obtained at the Lubango, Mongu and Ascension Island AERONET stations (daytime model outputs only) for September 2016.










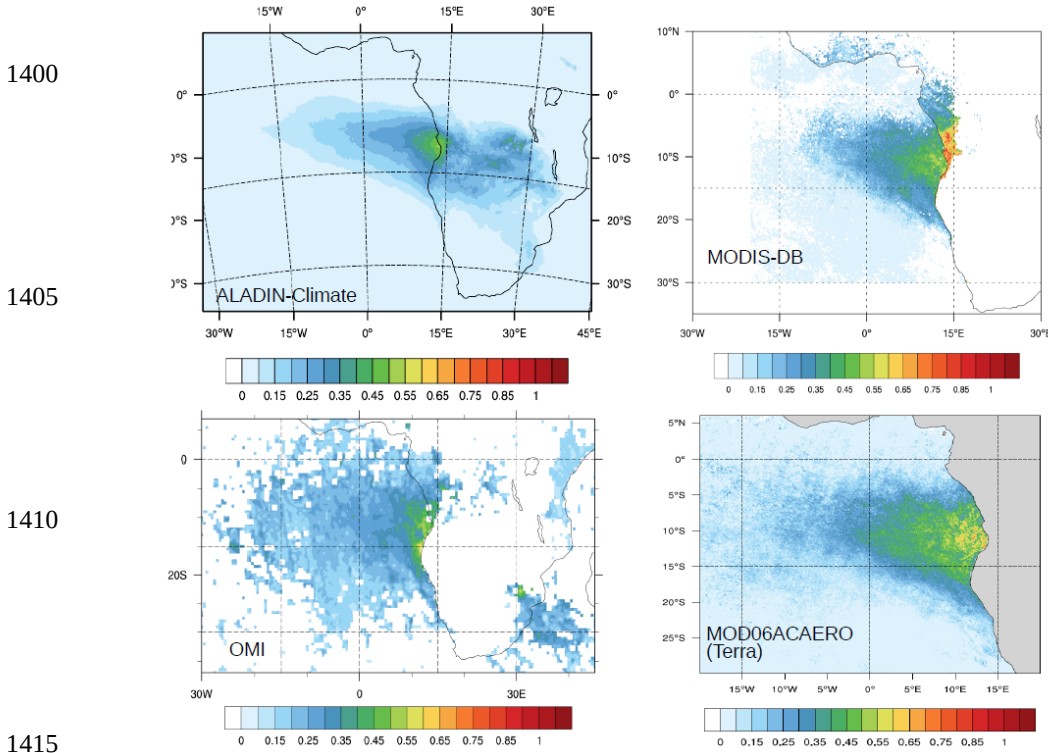



**Figure 6** Total monthly-averaged ACAOD (at 550 nm) for September 2016 simulated by the ALADIN-Climate model and derived from MODIS (Deep-Blue and Meyer retrievals) and OMI instruments.






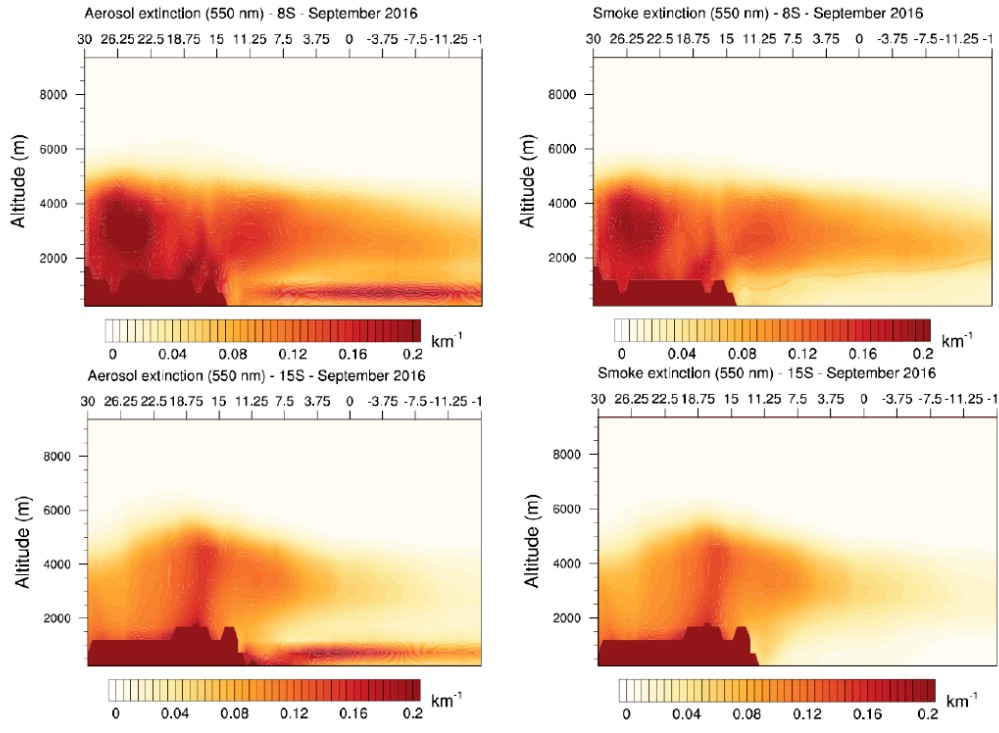

**Figure 7.** Monthly averaged vertical profiles of the total aerosol (left) and smoke (right) extinction
coefficient (at 550 nm) simulated by the ALADIN-Climate model for two transects at latitudes of 8
and 15°S. Longitudes are between 30°E to 11.5°W.













**Figure 8.** Vertical profiles of aerosol extinction coefficient (in km$^{-1}$) at three different wavelenghts
from ALADIN-Climate and HSRL-2 instrument (red: 1064 nm / purple: 550 nm / blue: 440 nm),
associated with the mean values (note that the wavelenghts are not exactly similar, especially in
ultraviolet). Also reported are the vertical profiles of RH simulated by ALADIN-Climat (dotted
black) and MERRA2 (black), as well as the ALADIN-Climate cloud fraction (dotted yelow).




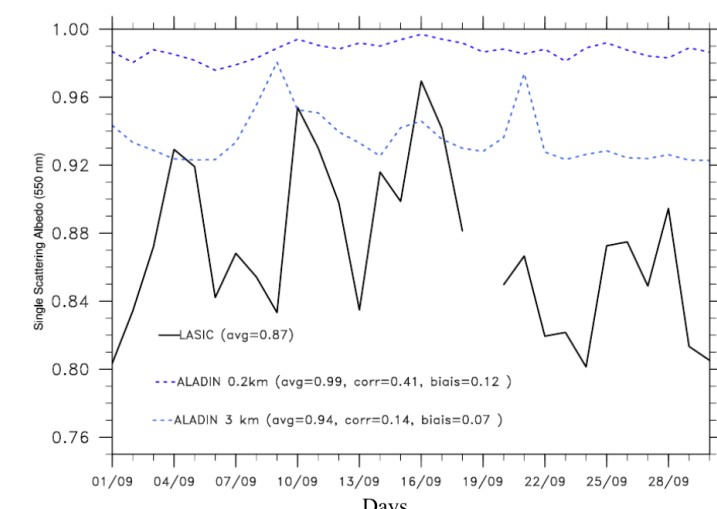



**Figure 9.** Daily-mean in-situ SSA estimated at the surface at Ascension Island (LASIC) and simulated with the ALADIN-Climate model at two different altitudes (0.2 and 3 km).










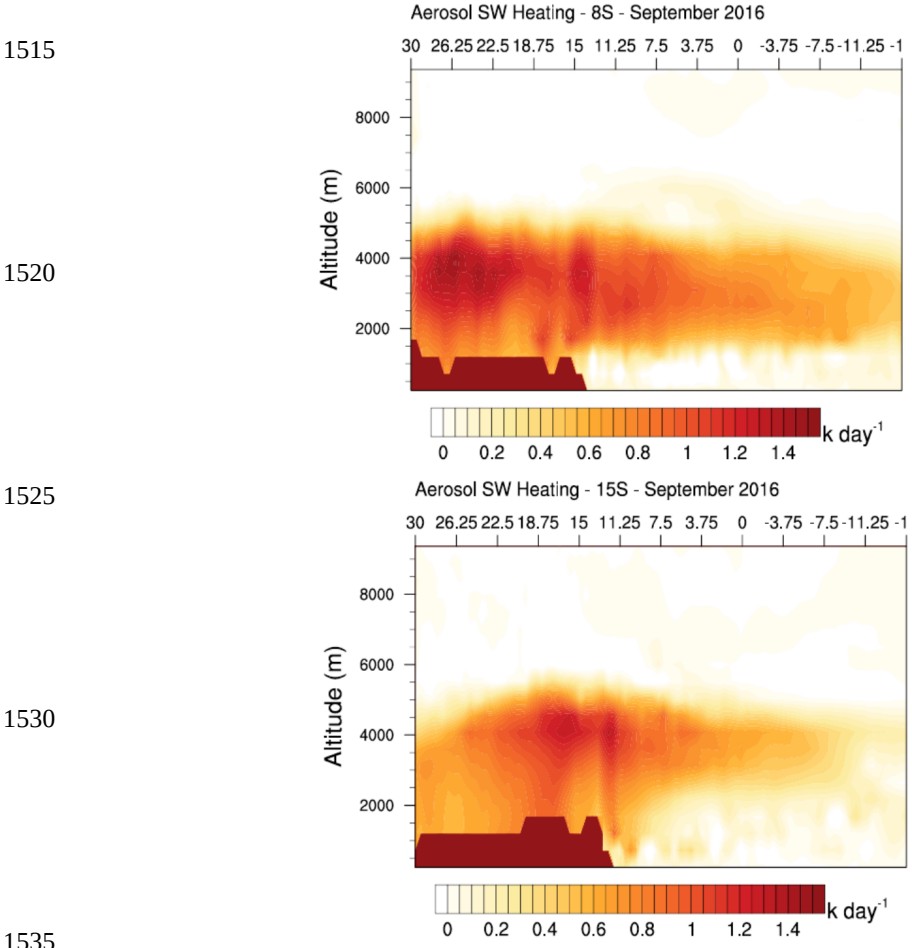





**Figure 10.** Monthly-mean (September 2016) aerosol SW heating rate vertical profiles simulated by the ALADIN-Climate model for two transects at latitudes of 8 (top) and 15°S (bottom). Longitudes are between 30°E to 11.5°W.







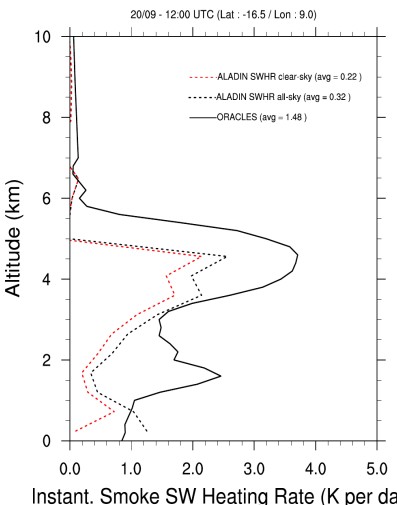

**Figure 11.** Instantaneous SW heating rate (12:00 UTC) only due to smoke aerosols, obtained from
ORACLES aircraft data and simulated by the ALADIN-Climate model in clear-sky (red dashed)
and all-sky (blue dashed) conditions.









**Figure 12.** Vertical profiles of aerosol extinction coefficient (at 550 nm) obtained from ALADIN-Climate for the SMK and SMK_SN simulations, and derived from the HSRL-2 instrument for the 24/09, associated to the vertical mean (left pannels). Also reported the RH obtained from the same simulations and MERRA2 data (right pannels).






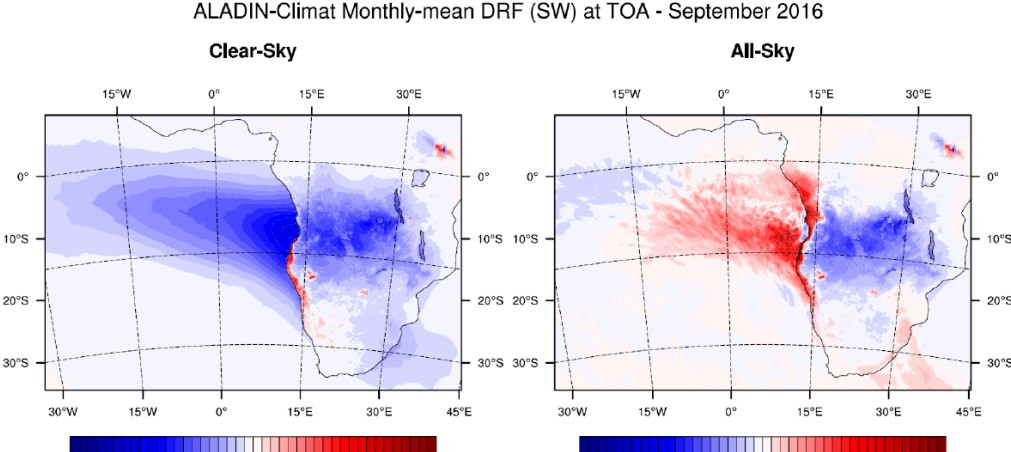

**Figure 13.** Monthly-mean SW DRF (W.m$^{-2}$) exerted by smoke particles at TOA for the September 2016 period in clear-sky (left) and all-sky (right) conditions.









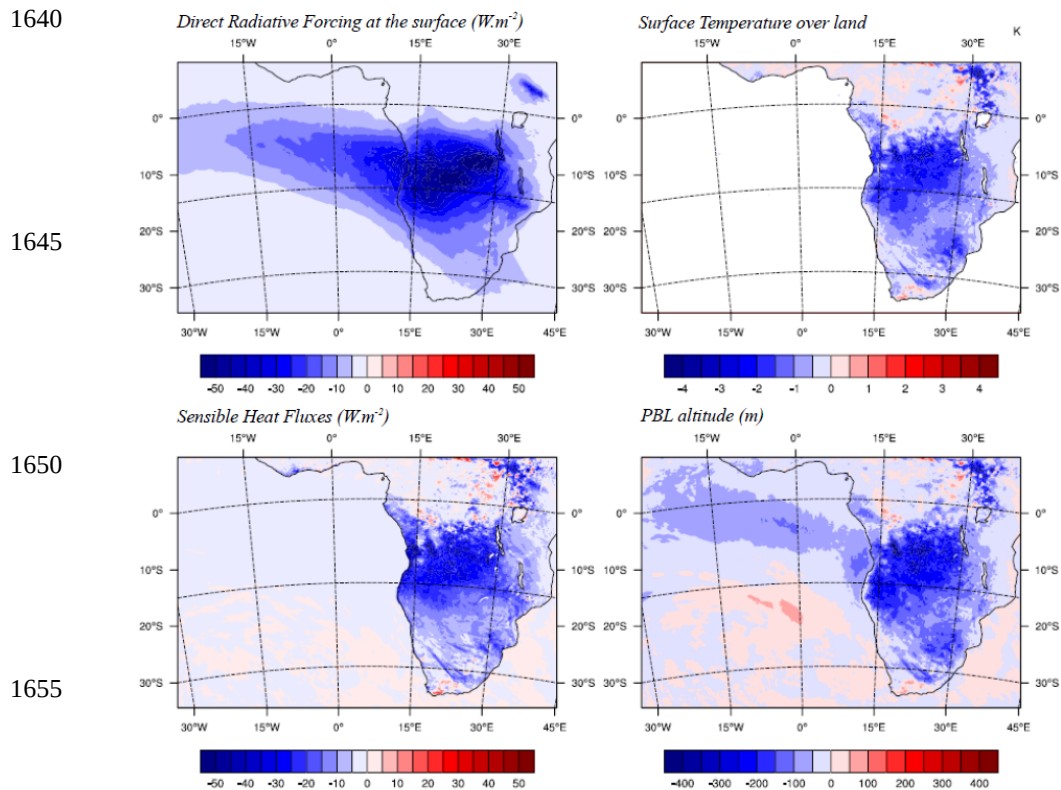



**Figure 14.** Differences between the CTL and SMK ALADIN-Climate runs in the monthly-mean
(September 2016) SW surface radiations (top left), 2 meter continental temperature (top right),
sensible heat fluxes (bottom left) and PBL height (bottom right).



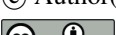









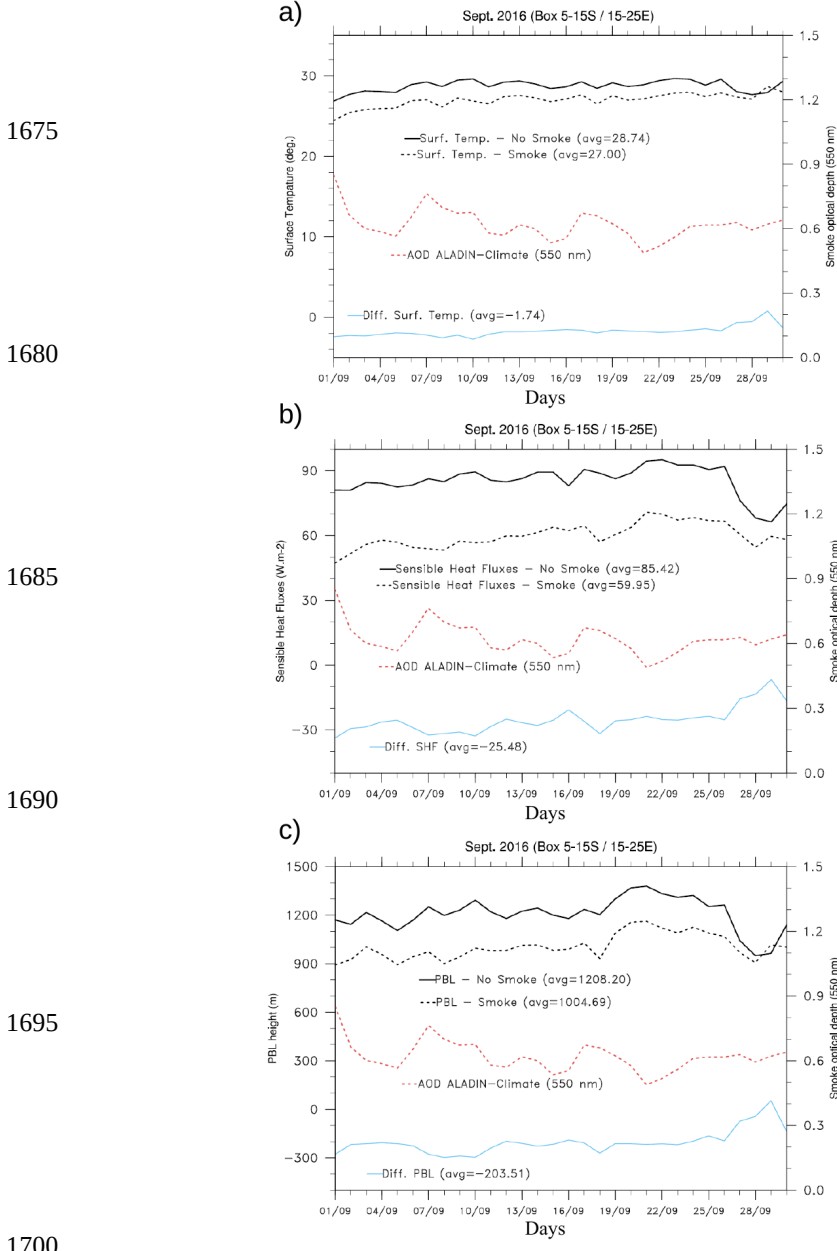

**Figure 15.** Daily-mean Surface Temperature, Sensible Heat Fluxes and PBL height obtained from the CTL (black lines) and SMK (dashed black) ALADIN-Climate simulations for september 2016. The AOD (red dashed lines) and the difference between the two simulations (CTL and SMK) for each variables are also reported (blue lines).