# Peer review of "Simulation of the transport, vertical distribution, optical properties and radiative impact of smoke aerosols with the ALADIN regional climate model during the ORACLES-2016 and LASIC experiments."

_Atmospheric Chemistry and Physics, 2018_

## Referee Comment (RC1) · Reid (Referee) · 3 Jan 2019

This is a well written and straightforward paper on model simulations of heating rates compared to and at times constrained by the ORACLES and LASIC field campaigns. I found it easy to ready and well laid out. They take a best available run with the model, and clearly spell out modeled radiative, temperature and PBL effects. They note biases and deficiencies, particularly in refer-
ence to wo and smoke vertical profile. I have a few minor comments (listed below) but one major comment. I would like to direct the authors to Tom Ecks paper https://agupubs.onlinelibrary.wiley.com/doi/full/10.1002/jgrd.50500 on the seasonal trends of wo over Africa. One thing the paper failed to account for is that wo over Africa has a very strong and very predictable trend due to a systematic shift in grass burning in the early season to more wooded fire late in the season, roughly 0.83 in early July, to 0.92 in mid-October at 440 nm. Yet, the real part of the index of refraction and the size distribution remain relatively static. This makes for an outstanding natural partial derivative on the sensitivity of the system to black carbon. However in the paper the black carbon mass fraction is static for the burning season. While I do not think that they necessarily need to do another run (as the model simulation is for a the middle 2 months and results are largely aggregated), I think that at least a paragraph or two needs to be present adding context to their run and providing rough error estimates, sensitivity and implication (if any) of this strong seasonal trend. In particular, please compare this finding to what you found in the model (Line 618).

I have lots minor comments that I think might clarify the paper. Some of this is because it is just the way the model was constructed and the investigators are sort of stuck with it. I don't mind so much of their assumed parameters are out of expected range by a little bit, but it should probably be noted. Also, things that seem minor information is actually very helpful later on when people try to reconcile model runs and observations. So please do your best to address these

Line 79: the first few times please state worming/cooling in association with positive and negative DRF for clarity for readers

Line 163. The use of OC/BC as biomass burning tracer with fixed microphysical and optical properties and basically being in the same emission category with anthropogenic is somewhat problematic and their hypothesis 'implications" are almost certainly violated in the study regime, especially on the northern end of the core biomass burning feature. This is a recurring problem in the modeling community, and has led to significant discussion within the ICAP community. The bottom line is that carbonations species are fundamentally different from biomass burning and anthropogenic/biogenic sources, and should be treated separately in models. But, model architecture is not so easy to change. I think the authors need to be clear about this up front and add a few lines discussing specifically what this does to the simulation. Fortunately for them, biomass burning particle evolution tends to be rather fast, slowing down by the time it reaches the coastline (https://www.atmos-chem-phys.net/5/799/2005/ ) This said, however, African smoke has shown evidence of evaporation/sublimation as noted in https://link.springer.com/article/10.1007/BF00708178 Line 171. Again, based on lit review https://www.atmos-chem-phys.net/5/799/2005/ is more in line with Vakkari. It is complicated because one has to decide what the initial state is to start the clock ticking. There is substantial evidence of difference in smoke properties from the base and top of a smoke column too. I think this is fairly moot though given the large scale nature of the simulation. Line 188. Just an FYI, you should note that these values of MEE are just on the upper half of what has been gravimetrically observed https://www.atmos-chem-phys.net/5/827/2005/acp-5-827-2005.pdf But 5 is a nice round number.

Line 242: See comment on line 171

Line 247: this ratio is also a bit high. Consider, OC makes up about 40-50% of mass, so a ratio of 2.3 makes over 100% of mass, and we know that Africa smoke is dominated by grass fires which have a high inorganic fraction.

Line 284: I think the site is now Mongu Inn instead of Mongu

Line 293: Which AERONET version was used? V3 came on line recently so it is not obvious.

Line 326: What was the altitude above clouds the reflectance was taken at?

Line 554; CALIOP is all CAPS

Line 627: How much do you think assumptions of hygroscopicity versus speciation

plays into this? Granted, this is mostly an absence of BC in smoke, but you still may have a factor of 2 floating around given the high RH in the MBL.

Line 635: can you elaborate here? Paragraph around Line 683: This paragraph is bordering on a non-sequitur. Wv is a great tracer, but is fundamentally different from RH. So careful how you talk about humidity and optical properties. Paragraph around 695: Can you compare model versus measured f(RH) directly here? Paragraph starting 760: On PBL impact: Please be specific what you mean by PBL height, are you referring to the actual top of the PBL (that can be somewhat amorphous given the depth of the entertainment zone in cloud atmospheres, but comes out often as a hazy model metric) or are you referring specifically to the top of the mixed layer? If you are referring to a systematic change in the base of the inversion, please state that clearly throughout. Also, Just curious, any wind impacts? Mark Jacobson years ago was reporting big wind impacts in global GATOR simulations. See any evidence? Regardless positive or negative it is worth mentioning any notable wind impacts. Also, can you calculate a specific surface temperature change per unit optical depth? See this for comparison https://www.atmos-chem-phys.net/16/6475/2016/

---

## Referee Comment (RC2) · Anonymous Referee #2 · 7 Jan 2019

Review of "Simulation of the transport, vertical distribution, optical properties and radiative impact of smoke aerosols with the ALADIN regional climate model during the ORACLES-2016 and LASIC experiments" by Mallet et al., submitted to *Atmos. Chem. Phys.*

In this study, the authors compare a simulation of stratocumulus clouds and biomass-burning aerosols over the southeastern Atlantic to aircraft and satellite retrievals of

clouds and aerosol properties. They find that the simulation is satisfactory in the first order, although aerosol extinction and absorption and cloud fraction are underestimated, and cloud optical thickness is overestimated. A simulation nudged to reanalysis outperforms the free-running model because nudging improves simulated relative humidity, which in turns improves aerosol extinction through hygroscopicity.

The paper is interesting and well-written. Many aspects of modelled aerosols and clouds relevant to the direct radiative effects of biomass-burning aerosols are evaluated against multiple observational datasets. The discussion is convincing and supported by a large number of figures.

I have only one main comment: the authors should clearly set expectations in section 3, by which I mean to state clearly what the model should be capable of in terms of reproducing spatial and temporal variability, and what comparison the satellite products are able to usefully provide. The reason it matters is that the model seems to be using monthly-averaged emissions, which may not even be for the year 2016. So the model cannot be expected to reproduce daily distributions, even when nudged. In addition, temporal sampling of satellite retrievals is limited, as stated by the authors, which means that model and observations should really be co-located temporally before being compared (Schutgens et al. doi:10.5194/acp-16-1065-2016 2016). This is not done here consistently, so there the comparisons can only be qualitative. In addition, I note that it is becoming more challenging to avoid circular reasoning, i.e. not comparing models using satellite-derived emissions to the same satellite products, and to reanalyses that assimilate some of those same products. This is not a criticism of the paper, but it suggests that having multiple, independent observational datasets for the same variables is becoming increasingly important.

Addressing my main comment, and the other comments below, should represent minor revisions.

Other comments:

- Line 63: It would be good to define DRF here: with respect to no-aerosols. Note that the IPCC calls that direct radiative *effect* (DRE). *Forcing* is when defined with respect to pre-industrial aerosols.

- Line 65: the "well-known" cooling effect is only true on a global average, so there is no contradiction really. I suggest rephrasing to contrast the top-of-atmosphere radiative effect of scattering and absorbing aerosols.

- Line 78: Which domain?

- Lines 111–112: What are the roles of AEROCLO and CLARIFY in this paper?

- Line 142: It would make more sense to start the chain of processes with surface emissions.

- Line 160: "also represented" – I suppose that semi-direct effects do not have a dedicated representation in ALADIN. They implicitly derive from direct effects.

- Lines 164–165: Strictly speaking, smoke is often anthropogenic – it is just that emission people cannot tell the two components apart so call the dataset "biomass-burning".

- Line 169: Is the fresh mode hygrophobic?

- Line 172: "more aged" is unclear. Once in the aged mode, aerosols cannot aged further in the model. Or are you saying that an e-folding time of 3 or 6 hours won't make much difference for SAO properties?

- Line 183: Is that the mean over the year or over the biomass-burning season? The latter would make more sense.

- Line 227: "forced-mode configuration" is ambiguous – does that mean fixed SSTs?

- Line 229: Need more information about those CMIP6 emissions, including a reference. This is a crucial aspect of the model, which will influence its capabilities and the interpretation of the comparison to observations. Is the dataset GFED- or GFAS- (i.e. satellite-)based? Are emissions really for the given year or just interpolations between key years, liked they did in CMIP5?

- Line 234: The boundary layer is probably deeper than just the first model level.

- Line 235: The Dentener recommendations are more complex that just injecting into the first model level. See their Table 2.

- Line 244: Is the ratio applied to the emissions or when mass is transferred from the fresh to the aged mode?

- Line 253: Worth noting that 0.15 represents about 20% of BBA AOD, so not a small change.

- Lines 298–302: Why is that a good thing for aerosol retrievals? Better correction of the Rayleigh contribution?

- Line 364: "ice clouds are not processed" is unclear. Does that mean that scenes containing ice clouds are discarded completely?

- Line 386: Does that retrieval suffers from the issues raised by Haywood et al. doi:10.1256/qj.03.100 (2004)? If so, that is a problem for the present study.

- Lines 395–396: Suggests shortening the title to "Reanalyses of atmospheric composition".

- Line 424: But at this stage of the analysis, it is not yet known that clouds are too bright – it will be shown in the following section.

- Line 431: What kind of parameterizations are they?

- Lines 441–442: It would be worth noting that indirect effects are relevant to DRF, because DRF depends on the albedo of the underlying stratocumulus.

- Line 448: Note that the CAMS Reanalysis, successor to MACC, covers 2016, so could be added to the comparison. See https://apps.ecmwf.int/data-catalogues/cams-reanalysis/?class=mc&expver=eac4

- Lines 455, 477, and 482: Those large differences are surprising because MERRA is supposed to be assimilating MODIS! Perhaps a different collection? The fact that MERRA assimilates MODIS should explain the good temporal correlation, though.

- Lines 458–472: I agree that land-ocean contrast in satellite products are worth investigating further. At first, I though that marine aerosols could possibly explain why there is more AOD over ocean than over land. But if we assume that the contrast observed on Figure 3 south of the BBA plume, say 20S, is only due to seasalt, we only get about +0.1 contrast. Reporting that to within the plume leaves about 0.1-0.2 of contrast unexplained.

- Line 462: MODIS products include uncertainties so it is a good place to use them. Perhaps show an uncertainty range on Figure 4?

- Line 472: "more robust" – in terms of sampling yes, but the AOD retrievals are also more uncertain over land than over ocean because the surface albedo is larger.

- line 509: How is ACAOD calculated in the model? It is not always easy to determine where the cloud top is.

- Lines 523–524: What did Shinozuka et al. find?

- Lines 531–539: That analysis supports the idea that injection heights are not that important. Aerosols are lofted by convection anyway.

- Line 552: Is the decrease in extinction driven by a decrease in mass?

- Line 562: The statement on advection contradicts line 141. I fail to see why the model could not represent those BBA incursions into the BL – it might be that the model of Gordon et al. is wrong!

- Line 587–589: So the RH biases go in the right direction to (partly) explain the extinction biases.

- Line 615: The agreement is good but observational uncertainties are large.

- Lines 624–632: That paragraph is confusing. Is the comparison fair? Is the model simulating BBA on the days of the comparison? Can we be sure that LASIC is observing transported BBA and not local sources?

- Line 633: "reflect" – the observations are insufficient to link that absorption to ageing during transport. I am not convinced the model is wrong here.

- Line 650: Are all those studies based on modelling?

- Line 674: Section 4.3 is interesting. Essentially aerosol DRF errors in the SAO are driven by non-aerosol aspects. It is however unclear if the increased water vapour is due to the fires or because of transport in convective air masses. I suppose it is the latter, since the model does not emit water vapour with fires, nor does it account for additional buoyancy from the fires. Although lines 722–723 are ambiguous about what the model really does.

- Line 554: "we suspect". We were promised a bit more. Can we have an integrated assessment of what the different model biases in CF, COD, ACAOD, and SSA mean for DRF?

- Table 2: It would be useful to add a column listing the periods covered by each product.

Technical comments:

- Line 105: Consist to -> is to

- Line 187: $g$ has not yet been defined, unless I missed it.

- Line 221: The definition of the domain encompasses the main biomass-burning sources *of that region*, and also the transport to the Atlantic ocean.

- Line 224: Suggest moving the Mlawer reference to line 155 for consistency with FMR.

- Line 234: Not sure "accordingly" is the right word here.

- Line 241: produce -> produced

- Line 357: CER has not yet been defined.

- Figure 7: Could the orography be put in a colour that is not in the colour scale used for aerosol extinction? Grey perhaps?
* * *

---

## Author Comment (AC1) · 1 Mar 2019

*Dear Editor,*

*We first would like to thank the reviewer for all the remarks and suggestions that we used to improve the manuscript. We have tried to take into account most of the mentioned points.*

Review of "Simulation of the transport, vertical distribution, optical properties and radiative impact of smoke aerosols with the ALADIN regional climate model during the ORACLES-2016 and LASIC experiments" by Mallet et al., submitted to Atmos. Chem. Phys.

In this study, the authors compare a simulation of stratocumulus clouds and biomass-burning aerosols over the southeastern Atlantic to aircraft and satellite retrievals of clouds and aerosol properties. They find that the simulation is satisfactory in the first order, although aerosol extinction and absorption and cloud fraction are underestimated, and cloud optical thickness is overestimated. A simulation nudged to reanalysis outperforms the free-running model because nudging improves simulated relative humidity, which in turns improves aerosol extinction through hygroscopicity.

The paper is interesting and well-written. Many aspects of modelled aerosols and clouds relevant to the direct radiative effects of biomass-burning aerosols are evaluated against multiple observational datasets. The discussion is convincing and supported by a large number of figures.

I have only one main comment: the authors should clearly set expectations in section 3, by which I mean to state clearly what the model should be capable of in terms of reproducing spatial and temporal variability, and what comparison the satellite products are able to usefully provide. The reason it matters is that the model seems to be using monthly-averaged emissions, which may not even be for the year 2016. So the model cannot be expected to reproduce daily distributions, even when nudged. In addition, temporal sampling of satellite retrievals is limited, as stated by the authors, which means that model and observations should really be co-located temporally before being compared (Schutgens et al. doi:10.5194/acp-16-1065-2016 2016). This is not done here consistently, so there the comparisons can only be qualitative. In addition, I note that it is becoming more challenging to avoid circular reasoning, i.e. not comparing models using satellite-derived emissions to the same satellite products, and to reanalyses that assimilate some of those same products. This is not a criticism of the paper, but it suggests that having multiple, independent observational datasets for the same variables is becoming increasingly important.

*This is effectively right, and represents an important limitation for the comparisons between models and satellite observations. As remarked by the reviewer and concerning the emissions, we have now clearly stated in the new version (section 3.1) the methodology used in this study and what the model is able to reproduce. We especially point out the fact that comparisons are only « qualititive ». Indeed, the ALADIN-Climate simulation used « monthly-mean » biomass burning historical CMIP6 emissions, based on historical GFED emission. The reference which presents these CMIP6 emissions (van Marle, et al., 2017) is now provided in the new version. The main point of the methodology ; namely the use of the Global Fire Emissions Database version 4 (GFED4s) for the 1997–2015 period, is now indicated in the text.*

*In this work, the ALADIN-Climate simulation used the « monthly-mean » emission for one of the closest year of the historical CMIP6 emissions; namely the year 2014, as 2016 is not avalaible. This methodology implies « realistic » emissions in terms of magnitude but does not represent daily variations. This important point is now detailed and discussed in the different parts (sections 3.1 and 4.2.1 notably) to underline our methodology and clearly point that this does not allow the model to reproduce precisely daily aerosol distributions over SAO.*

*It should be noted here that our main motivation was to use the ALADIN-Climate model in its « climate» configuration to study its ability at reproducing the main cloud and aerosol optical properties over the SAO. This represents a crucial and necessary step before using such regional climate model in an exactly similar configuration at climatic scales to address the radiative and climate impact of BBA over Southern Africa, which clearly represents the main vocation of such a modeling tool, compared to meso-scale fine resolution models (such as WRF-C). We remind this important aspect in the new version (section 3.1).*

*In parallel, we have checked that the BC+OC AOD for September 2014 was found to be consistent, in terms of AOD, to CAMS climatology for the 2008-2015 period. The 2014 september anomaly (see the following figure now provided in Appendix) indicates small differences ~0.05-0.10 over the BBA sources for September 2014 compared to the climatology. Additionally, Sayer et al. (2019) found similar above-cloud and total-column AOD over the southern Atlantic Ocean in 2014 and 2016. This would suggest that the effect of potential differences between 2014 and 2016 for BC-OC emissions over the biomass burning region is likely to be small. These points are now underlined in the part 3.1 of the new version. The following reference has been added.*

*Sayer et al., Two decades observing smoke above clouds in the south-eastern Atlantic Ocean : Deep Blue algorithm updates and validation with ORACLES field campaign data, Atmos. Meas. Tech. Discussion, in review, 2019.*

[Figure]

*New Figure S1. September 2014 BC-OC AOD anomaly (left) compared to the 2008-2015 period (September month only) and the total mean BC+OC AOD for the 2008-2015 period (right) from CAMS reanalyses. This figure is now provided in the new version of the Appendix.*

*Concerning the comparisons with satellite data, we have also mentioned in the new version the uncertainties/limitations related to the use of monthly-mean emissions to force the model. In addition, all the comparisons between the ALADIN-Climate model and satellite retrievals have been performed by using only ALADIN-Climate outputs at the different satellites (OMI, MODIS) over-passing (i.e., between 10:30 & 13:00 UTC). This is now indicated in the new version and clearly stated for the figures 3 and 4. In parallel, the figure 6 has been modified using MODIS and OMI equator crossing times (10:30 and 13:30 UTC) for the ALADIN-Climat outputs. This leads to moderate changes in the simulated ACAOD (small decrease) over the continent.*

Addressing my main comment, and the other comments below, should represent minor revisions.

Other comments:

Line 63: It would be good to define DRF here: with respect to no-aerosols. Note that the IPCC calls that direct radiative effect (DRE). Forcing is when defined with respect to pre-industrial aerosols.
**This is right and now mentioned in the new version.**

Line 65: the "well-known" cooling effect is only true on a global average, so there is no contradiction really. I suggest rephrasing to contrast the top-of-atmosphere radiative effect of scattering and absorbing aerosols.
**This is right and changed in the text.**

Line 78: Which domain?

*The domain (4–18°S, 5°W–14°E) is now defined.*

Lines 111–112: What are the roles of AEROCLO and CLARIFY in this paper ?
*This was only a point of the general context, indicating the different programs conducted over this region with co-incident experimental campaigns. We think that this is of interest to mention in the introduction but we can remove it if necessary. We have clearly stated that only the data from LASIC and ORACLES programs have been used in this work.*

Line 142: It would make more sense to start the chain of processes with surface emissions.
*This is now modified in the new version.*

Line 160: "also represented" – I suppose that semi-direct effects do not have a dedicated representation in ALADIN. They implicitly derive from direct effects.
*This is right and we have now replaced « also represented » by «is derived from the direct effect ».*

Lines 164–165: Strictly speaking, smoke is often anthropogenic – it is just that emission people cannot tell the two components apart so call the dataset "biomass-burning".
*This is noted. We prefer to keep the term « biomass-burning » in the text to avoid confusions with BC and OC emissions from anthropogenic activity.*

Line 169: Is the fresh mode hygrophobic?
*Yes. This specific mode has lesser hygroscopic properties than aged smoke, as reported in the relation 1 and Table 2. This point is now precised.*

Line 172: "more aged" is unclear. Once in the aged mode, aerosols cannot aged further in the model. Or are you saying that an e-folding time of 3 or 6 hours won't make much difference for SAO properties ?
*We rewrote this sentence : « The smoke over the SAO is expected to have aged by 5-7 days...».*

Line 183: Is that the mean over the year or over the biomass-burning season? The latter would make more sense.
*This is the mean value over the year, and further developments are needed to better take into account the seasonal variations of smoke optical properties (especially from grass to forest burning). This clearly represents one limitation and following this remark (as also noted by the reviewer 1), we have added a new simulation, SMK_SSA using SSA of 0.92 to test the sensitivity of BBA absorbing properties on SW heating, direct forcing, surface temperature as well as the PBL development. We have now indicated the additional results in the new Figure 13 and Figure S7 (provided in Appendix). In parallel, the results are also summarized in a new Table (Table 3).*

*In terms of SW radiative heating and direct radiative forcing, this additional simulation indicates :*

*- a significant decrease of the SW radiative heating induced by smoke aerosols. For example and over the Box_S (defined over the sources of biomass burning), SW heating at 3km is passing from +1.15°K by day to + 0.58°K by day. This result moderates the SW heating induced by smoke at the end of the biomass burning season. This specific point is now included in the article (paragraph 4.2.4.3) and in the new Table 3,*

*- a significant change in the monthly-mean (September 2016) DRF at TOA, passing from a positive (+4.2 W.m$^{-2}$) to negative (-0.54 W.m$^{-2}$) direct forcing (see new Figure 13 and Table 3) over the Box_O (ocean). This means that the positive direct forcing at TOA could be lesser in intensity at the end of the BBA season (late october). This important point is now mentioned in the part 5.1 and the results of the new ALADIN-Climate (SMK_SSA) simulation including more scattering BBA (SSA of 0.92 at 550 nm) are included in the Table 3,*

*- a more intense negative DRF at TOA over smoke sources due to more scattering BBA. Over the box_S, the monthly-mean value (Table 3) is increasing from -3.9 W.m$^{-2}$ (SMK) to -7.3 W.m$^{-2}$ (SMK_SSA). This specific point is added in the disussions and in the Table 3,*

*- a less pronounced positive DRF at TOA along the Southern African coast and Gabon due to more scattering BBA (now indicated in the paragraph 5.1),*

[Figure]

*New figure 13 including (bottom) the monthly-mean DRF exerted at TOA for more scattering smoke aerosols (SMK_SSA ALADIN simulation).*

*Concerning the impact of BBA on other variables (SW radiations at the surface, surface temperature, sensible heat fluxes and PBL height), we did exactly the same figure as Figure 14 but for the new simulation (SSA_SMK). This new figure has been added in Supplement material (S8) and the main results are discussed in the Part 5.2. We have notably added these clarification's :*

*« Finally, the comparisons with the SMK_SSA simulations (not shown, Figure S8) indicate a decrease of the surface radiative forcing both over continent and ocean. As reported in Table 3, the monthly-mean DRF at BOA is about -39 W.m$^{-2}$ and -25 W.m$^{-2}$ over Box_S (biomasse burning sources) for the SMK and SMK_SSA simulations, respectively. The same result is obtained over SAO. This is due to the decrease of SW radiations absorbed by smoke in the SMK_SSA simulation, increasing the SW radiations reaching the surface. This could be also due, to a lesser extent, to some changes in aerosol loading due to modifications in the dynamics and precipitation between the two simulations. This induces a less pronounced impact of BBA on the surface temperature and sensible heat fluxes in the SMK_SSA run. The increase of SW surface radiations, associated to low absorption by BBA in SMK_SSA, decrease the impact of smoke on the PBL development (Figure S8). As mentioned previsouly, these results suggest that the impact of BBA on the surface fluxes and dynamics are certainly slightly lower at the end of the biomass burning season. »*

Line 227: "forced-mode configuration" is ambiguous – does that mean fixed SSTs ?
*This is effectively the case. This point is now clearly mentioned.*

Line 229: Need more information about those CMIP6 emissions, including a reference. This is a crucial aspect of the model, which will influence its capabilities and the interpretation of the comparison to observations. Is the dataset GFED- or GFAS- (i.e. satellite-)based? Are emissions really for the given year or just interpolations between key years, liked they did in CMIP5?

*As mentioned above, the biomass-burning emission used in the model are monthly-mean values based on GFED inventory database. The reference (van Marle, et al., 2017) describing the methodology to build CMIP6 emission are now indicated in the new version. CMIP6 emissions are developped with realistic-timing emission for the 1997-2015 period but not for 2016. This point is now clearly mentioned in the new version associated with the related limitations.*

*van Marle, M. J. E., Kloster, S., Magi, B. I., Marlon, J. R., Daniau, A.-L., Field, R. D., Arneth, A., Forrest, M., Hantson, S., Kehrwald, N. M., Knorr, W., Lasslop, G., Li, F., Mangeon, S., Yue, C., Kaiser, J. W., and van der Werf, G. R.: Historic global biomass burning emissions for CMIP6 (BB4CMIP) based on merging satellite observations with proxies and fire models (1750–2015), Geosci. Model Dev., 10, 3329-3357, https://doi.org/10.5194/gmd-10-3329-2017, 2017.*

Line 234: The boundary layer is probably deeper than just the first model level.
*This is right and our sentence is confusing. We would like to indicate near the main biomass burning sources and not the surface layer. This sentence is now rephrased in that sense.*

Line 235: The Dentener recommendations are more complex that just injecting into the first model level. See their Table 2.
*This is right. This point is now modified and we have now removed the following sentence : « smoke emissions force the model at the first model level following the recommendations from the first phase of AEROCOM ».*

Line 244: Is the ratio applied to the emissions or when mass is transferred from the fresh to the aged mode ?
*This is done when the mass is transferred from the fresh to aged mode in the model. This point is now stated.*

Line 253: Worth noting that 0.15 represents about 20% of BBA AOD, so not a small change.
*This is right and we have now rephrased the sentence to indicate that changes coud be important for smoke AOD.*

Lines 298–302: Why is that a good thing for aerosol retrievals? Better correction of the Rayleigh contribution?
*To clarify how this helps aerosol retrievals, we have edited the sentence like this: "…but not aerosol scattering. This channel therefore provides a direct observation of the attenuation of the signal and allows for the direct calculation of aerosol extinction without external constraints or assumptions on the aerosol optical depth or lidar ratio."*

Line 364: "ice clouds are not processed" is unclear. Does that mean that scenes containing ice clouds are discarded completely?
*This is effectively the case and the scenes containing ice clouds are discarded completely. It should be noted that the different boxes used to evaluate the model are characterized by negligible high cloud fraction.*

Line 386: Does that retrieval suffers from the issues raised by Haywood et al. doi:10.1256/qj.03.100 (2004)? If so, that is a problem for the present study.
*This is a good remark and the possible effect of the presence of BBA on the stratocumulus properties retrievals has been studied and quantified recently by Seethala et al. (2018). This study indicates that, in the aerosol-affected months of July–August–September, SEVIRI LWP (based on the 1.6µm Cloud Effetive Radius) is biased by ~16%. This point is now clearly indicated in the new version and the following reference has been added.*

*Seethala, C., Meirink, J. F., Horváth, Á., Bennartz, R., and Roebeling, R.: Evaluating the diurnal cycle of South Atlantic stratocumulus clouds as observed by MSG SEVIRI, Atmos. Chem. Phys., 18, 13283-13304, https://doi.org/10.5194/acp-18-13283-2018, 2018.*

Lines 395–396: Suggests shortening the title to "Reanalyses of atmospheric composition".
***This is changed.***

Line 424: But at this stage of the analysis, it is not yet known that clouds are too bright – it will be shown in the following section.
***This is effectively right and we have now changed the sentence in the part 4.1.1.***

Line 431: What kind of parameterizations are they?
***The two parametrezations concern the calculation of the liquid cloud optical depth (COD) using LWP and Cloud effective radius but with different coefficients depending of the wavelenghts. This point is now detailled.***

Lines 441–442: It would be worth noting that indirect effects are relevant to DRF, because DRF depends on the albedo of the underlying stratocumulus.
***This is right and now added in the new version.***

Line 448: Note that the CAMS Reanalysis, successor to MACC, covers 2016, so could be added to the comparison. See https://apps.ecmwf.int/data-catalogues/ cams-reanalysis/?class=mc&expver=eac4
***This is an interesting point and the total AOD from CAMS reanalysis has been now added in the new Figure 3 and discussed in the text.***

Lines 455, 477, and 482: Those large differences are surprising because MERRA is supposed to be assimilating MODIS! Perhaps a different collection? The fact that MERRA assimilates MODIS should explain the good temporal correlation, though.
***To our knowledge, MERRA does not assimilate these MODIS products; it does a neural network-based retrieval on the MODIS radiances and assimilates the results of that instead. This leads to a better (compared to ALADIN-Climat) temporal correlation, which is now mentioned in the new version.***

Lines 458–472: I agree that land-ocean contrast in satellite products are worth investigating further. At first, I though that marine aerosols could possibly explain why there is more AOD over ocean than over land. But if we assume that the contrast observed on Figure 3 south of the BBA plume, say 20S, is only due to seasalt, we only get about +0.1 contrast. Reporting that to within the plume leaves about 0.1-0.2 of contrast unexplained.
***This is an interesting remark, which is now added in the text to explain part of the AOD contrast between land and ocean.***

Line 462: MODIS products include uncertainties so it is a good place to use them. Perhaps show an uncertainty range on Figure 4?
***This is now done using uncertainties provided by Sayer et al. (2016) and Gupta et al. (2018), for Deep Blue ACAOD and Dark Target AOD aerosol products, respectively.***

Line 472: "more robust" – in terms of sampling yes, but the AOD retrievals are also more uncertain over land than over ocean because the surface albedo is larger.
***This is right and the term « robust » is not appropriated and now removed.***

Line 509: How is ACAOD calculated in the model? It is not always easy to determine where the cloud top is.
***It is calculated as the integration from the model top to the cloud top. This point is now detailled.***

Lines 523–524: What did Shinozuka et al. Find?
***Shinozuka et al. indicate similar results than presented in this work, especially an understimate of the simulated ACAOD with the ALADIN-Climate model for different boxes defined over the SOA region. This specific point is now detailled in the new version.***

Lines 531–539: That analysis supports the idea that injection heights are not that important. Aerosols are lofted by convection anyway.
***This is right and this point is now added.***

Line 552: Is the decrease in extinction driven by a decrease in mass?
***The decrease in extinction is effectively mostly driven by a decrease in the BBA concentration. This is precised in the text.***

Line 562: The statement on advection contradicts line 141. I fail to see why the model could not represent those BBA incursions into the BL – it might be that the model of Gordon et al. is wrong!
***The term « advection » is not appropriated and we have now modified it. In addition, we have moderated this point indicating that the results obtained with ALADIN are different from those obtained by Gordon et al. (2018).***

Line 587–589: So the RH biases go in the right direction to (partly) explain the extinction biases.
***This is right and added in the new version.***

Line 615: The agreement is good but observational uncertainties are large.
***This is true and the uncertainties related to SSA AERONET retrievals are now indicated in the part 4.2.4.1.***

Lines 624–632: That paragraph is confusing. Is the comparison fair? Is the model simulating BBA on the days of the comparison? Can we be sure that LASIC is observing transported BBA and not local sources?
***The comparison has been done effectively for each days (daily mean) with the SSA observed at the surface at the ascension island ARM site. An important point is that the LASIC site is located on the remote windward side of the island and is not affected by local sources, of which there are few to begin with (no trash burning on the island). This is precised in the article.***

Line 633: "reflect" – the observations are insufficient to link that absorption to ageing during transport. I am not convinced the model is wrong here.
***This point is effectively not enough detailled. One indication that the aerosol sampled at the LASIC is aged is through the parameters f44 and f60, the fraction of the organic aerosol mass spectrum signal at m/z 44 and 60 respectively, in the data from the Aerosol Chemical Species Monitor (Alison Aiken, personal communication). The LASIC f44 and f60 values of approximately 0.2 and 0.002 respectively are characteristic of highly aged aerosols (Cubison et al., 2011). All these points are now included.***

***Cubison, M. J., Ortega, A. M., Hayes, P. L., Farmer, D. K., Day, D., Lechner, M. J., Brune, W. H., Apel, E., Diskin, G. S., Fisher, J. A., Fuelberg, H. E., Hecobian, A., Knapp, D. J., Mikoviny, T., Riemer, D., Sachse, G. W., Sessions, W., Weber, R. J., Weinheimer, A. J., Wisthaler, A., and Jimenez, J. L.: Effects of aging on organic aerosol from open biomass burning smoke in aircraft and laboratory studies, Atmos. Chem. Phys., 11, 12049-12064, https://doi.org/10.5194/acp-11-12049-2011, 2011.***

Line 650: Are all those studies based on modelling?
***Yes and this is now stated in the text.***

Line 674: Section 4.3 is interesting. Essentially aerosol DRF errors in the SAO are driven by non-aerosol aspects. It is however unclear if the increased water vapour is due to the fires or because of transport in convective air masses. I suppose it is the latter, since the model does not emit water vapour with fires, nor does it account for additional buoyancy from the fires. Although lines 722–723 are ambiguous about what the model really does.
***This is an open interesting question which is not resolvd at this time to our knowlegde. We agree with the reviewer that the presence of water vapor could be due at the first level to convective air masses as indicated by Adeyemi et al. (2015). In parallel, it is known that fires release important concentration of Wv but such processes are totally absent in the model at this time. This would represent interesting developments. This point has been detailled in the new version (3.1).***

Line 554: "we suspect". We were promised a bit more. Can we have an integrated assessment of what the different model biases in CF, COD, ACAOD, and SSA mean for DRF?

*This is an interesting remark, which represents an important work, notably by performing new sensitivity tests, based on additional simulations including variations of each variables (CF, COD, ACAOD and SSA) independantly. These new simulations/tests are unfortunatley outside the scope of the article.*

Table 2: It would be useful to add a column listing the periods covered by each product.
*This is now inserted in the new Table 2.*

Technical comments:

Line 105: Consist to -> is to
*This is modified.*

Line 187: g has not yet been defined, unless I missed it.
*This is effectively right and corrected in the new version.*

Line 221: The definition of the domain encompasses the main biomass-burning sources of that region, and also the transport to the Atlantic ocean.
*The ALADIN-Climate domain used to realize the simulation (Latitude : -37.1°S to 09.4°N ; Longitude :- 33.4°W to 45.4°E) is now cleary indicated in the Part 3.1.*

Line 224: Suggest moving the Mlawer reference to line 155 for consistency with FMR.
*This is changed.*

Line 234: Not sure "accordingly" is the right word here.
*This is right and « accordingly » has been changed by : « In the simulations, ... »*

Line 241: produce -> produced
*Now changed.*

Line 357: CER has not yet been defined.
*This is modified in the new version.*

Figure 7: Could the orography be put in a colour that is not in the colour scale used for aerosol extinction? Grey perhaps?
*This is changed in the new Figure 7.*

---

## Author Comment (AC2) · 1 Mar 2019

*Dear Editor,*

*We first would like to thank the reviewer for all the remarks and suggestions very useful to improve the manuscript. We have tried to take into account most of the mentionned following points.*

This is a well written and straightforward paper on model simulations of heating rates compared to and at times constrained by the ORACLES and LASIC field campaigns. I found it easy to ready and well laid out. They take a best available run with the model, and clearly spell out modeled radiative, temperature and PBL effects. They note biases and deficiencies, particularly in reference to wo and smoke vertical profile.

I have a few minor comments (listed below) but one major comment. I would like to direct the authors to Tom Ecks paper https://agupubs.onlinelibrary.wiley.com/doi/full/10.1002/jgrd.50500 on the seasonal trends of wo over Africa. One thing the paper failed to account for is that wo over Africa has a very strong and very predictable trend due to a systematic shift in grass burning in the early season to more wooded fire late in the season, roughly 0.83 in early July, to 0.92 in mid-October at 440 nm. Yet, the real part of the index of refraction and the size distribution remain relatively static. This makes for an outstanding natural partial derivative on the sensitivity of the system to black carbon. However in the paper the black carbon mass fraction is static for the burning season. While I do not think that they necessarily need to do another run (as the model simulation is for a the middle 2 months and results are largely aggregated), I think that at least a paragraph or two needs to be present adding context to their run and providing rough error estimates, sensitivity and implication (if any) of this strong seasonal trend. In particular, please compare this finding to what you found in the model (Line 618).

I have lots minor comments that I think might clarify the paper. Some of this is because it is just the way the model was constructed and the investigators are sort of stuck with it. I don't mind so much of their assumed parameters are out of expected range by a little bit, but it should probably be noted. Also, things that seem minor information is actually very helpful later on when people try to reconcile model runs and observations. So please do your best to address these

*We agree on this important limitation concerning the representation of smoke optical properties (notably absorption) in the ALADIN-Climate model. As fires are not explicitly resolved in the model, it is difficult to take into account changes in optical properties of smoke during the biomass burning season. In that sense, we have only investigated here the August-September period when smoke SSA remains low, around 0.84-0.86 at 500nm (AERONET retrievals ; Eck et al., 2013). In addition, this ALADIN-Climate simulation has been also constrained by recent in-situ observations (Zuidema et al., 2018) obtained within the marine boundary layer at Ascension island (SSA of ~0.80 at 550 nm in September 2016). Anyway, it is clear that this hypothesis is important and could have some implications on biomass burning shortwave (SW) heating rate and direct radiative effect especially for climate simulations including all the biomass buring season (from July to late october).*

*To address this specific point, we have now performed a new simulation, named SMK_SSA, which includes less aborbing smoke, more representative of the late season (September-October) as noted by Eck et al. (2013). In this sensitivity simulation, SSA has been fixed to 0.92 (550 nm) for smoke; the rest of BBA parameters being exactly similar. The description of this new simulation is now included in the paragraph 3.1 and this limitation of the ALADIN-Climate model is clearly reminded in the part 2.2. To illustrate these new results, the figure 13 has been modified (see the new Figure 13 below) and a new Table (Table 3) has been included.*

*Along the text, additional explanations are now provided in terms of sensitivity on (i) SW heating rate (paragraph 4.2.4.3), (ii) direct radiative effect exerted at the top of the atmosphere and (iii) the different impacts on surface SW radiations, temperature, sensible heat fluxes and PBL height.*

*In terms of SW radiative heating and direct radiative forcing, this new ALADIN simulation indicates :*

*- a significant decrease of the SW radiative heating induced by smoke. For example and over the Box_S (defined over the sources of biomass burning), SW heating at 3km is passing from +1.15°K by day to + 0.58°K by day. This point could moderate the heating induced by smoke at the end of the biomass burning season. This point is now included in the article (paragraph 4.2.4.3),*

*- a significant change in the monthly-mean (September 2016) DRF at TOA, passing from a positive (+4.2 W.m$^{-2}$) to negative (-0.54 W.m$^{-2}$) direct forcing (see new Figure 13 and Table 3) over the Box_O (ocean). This means that the positive direct forcing at TOA could be lesser in intensity at the end of the BBA season (late october). This important point is now mentioned in the part 5.1 and the results of the new ALADIN-Climate (SMK_SSA) simulation with more scattering smoke (SSA of 0.92) are included in the Table 3,*

*- a more intense negative DRF at TOA over smoke sources due to more scattering BBA. Over the box_S, the monthly-mean value DRF is increasing from -3.9 W.m$^{-2}$ (SMK) to -7.3 W.m$^{-2}$ (SMK_SSA). This specific point is added in the discussion. Values are reported in the Table 3,*

*- a less important positive DRF is observed at TOA along the Southern African coast and Gabon due to more scattering BBA. This important point is now indicated in the paragraph 5.1,*

[Figure]

*New figure 13 including (bottom) the monthly-mean DRF exerted at TOA for more scattering smoke (SMK_SSA simulation).*

*Concerning the impact of BBA on other variables (SW radiations at the surface, surface temperature, sensible heat fluxes and PBL height), we did exactly the same figure as Figure 14 but for the new simulation (SSA_SMK). This new figure (see below) has been added in Supplement material (S8) and the main results are discussed in the Part 5.2. We have notably added these clarifications :*

*«Finally, the comparisons with the SMK_SSA simulations (not shown, Figure S8) indicate a decrease of the surface radiative forcing both over continent and ocean. As reported in Table 3, the monthly-mean DRF at BOA is about -39 W.m$^{-2}$ and -25 W.m$^{-2}$ over Box_S (biomasse burning sources) for the SMK and SMK_SSA simulations, respectively. The same result is obtained over SAO. This is due to the decrease of SW radiations absorbed by smoke in the SMK_SSA simulation, increasing the SW radiations reaching the surface. This could be also due, to a lesser extent, to some changes in aerosol loading due to modifications in the dynamics and precipitation between the two simulations. This induces a less pronounced impact of BBA on the surface temperature and sensible heat fluxes in the SMK_SSA run. The increase of SW surface radiations, associated to low absorption by BBA in SMK_SSA, decrease the*

*impact of smoke on the PBL development (Figure S8). As mentioned previsouly, these results suggest that the impact of BBA on the surface fluxes and dynamics are certainly slightly lower at the end of the biomass burning season.»*

[Figure]

**The new Figure S7, which is now included in the supplement material.**

Line 79:  the first few times please state worming/cooling in association with positive and negative DRF for clarity for readers
***This point is now included in the new version.***

Line 163. The use of OC/BC as biomass burning tracer with fixed microphysical and optical properties and basically being in the same emission category with anthropogenic is somewhat problematic and their hypothesis 'implications" are almost certainly violated in the study regime, especially on the northern end of the core biomass burning feature.  This is a recurring problem in the modeling community, and has led to significant discussion within the ICAP community. The bottom line is that carbonations species are fundamentally different from biomass burning and anthropogenic/biogenic sources, and should be treated separately in models.  But, model architecture is not so easy to change.  I think the authors need to be clear about this up front and add a few lines discussing  specifically  what this does to the simulation. Fortunately for them, biomass burning particle evolution tends to be rather fast, slowing down by the time it reaches the coastline (https://www.atmos-chem-phys.net/5/799/2005/). This said, however, African smoke has shown evidence of evaporation/sublimation as noted in https://link.springer.com/article/10.1007/BF00708178.
***We agree with this remark, which represents one of the main motivations for including two new specific tracers in the ALADIN-Climate model to represent BBA, as presented in the part 2.2. This allows now to take into account specific properties for smoke particles, as the hygroscopic, e-folding time and optical properties. It allows notably to distinguish those particles from carbonaceous aerosols emitted from anthropogenic emissions with different properties. This point is now more detailed in the text (part 2.2).***

Line 171. Again, based on lit review https://www.atmos-chem-phys.net/5/799/2005/ is more in line with Vakkari. It is complicated because one has to decide what the initial state is to start the clock ticking. There is substantial evidence of difference in smoke properties from the base and top of a smoke column too. I think this is fairly moot though given the large scale nature of the simulation.

*This is right, and the reason why we had performed an additional simulation using a different efolding time (provided in the Appendix; Figure S1). We show that using a value of 3 hours (Vakkari et al., 2018) leads to change in AOD of about ~0.05 over the biomass burning region. This suggests, for this specific case, a modest impact on AOD compared to the hypothesis made on the POM to OC ratio. This point is indicated in the part 3.1 and deeper discussed in the new version.*

Line 188. Just an FYI, you should note that these values of MEE are just on the upper half of what has been gravimetrically observed https://www.atmos-chem-phys.net/5/827/2005/acp-5-827-2005.pdf But 5 is a nice round number.
*Thank you for this interesting review paper on smoke optical properties. We have incorporated it in the new version to discuss the values used in the ALADIN-Climat model.*

Line 242: See comment on line 171
*Please, see above concerning the use of a new simulation with different efolding time.*

Line 247: this ratio is also a bit high. Consider, OC makes up about 40-50% of mass, so a ratio of 2.3 makes over 100% of mass, and we know that Africa smoke is dominated by grass fires which have a high inorganic fraction.
*We agree on the fact that there are strong uncertainties on the POM to OC ratio. In this work, we have finally retained the value of Formenti et al. (2003). However and due to this large uncertainy, we have also conducted additional sensitiviy tests using two different values (2 and 3) of the POM to OC ratio. The results are presented in the Figure S1 (Appendix) showing an important sensitivity of ~0.2 on AOD over the box 5-15S/15-25E. This point is now more discussed in the new version.*

Line 284: I think the site is now Mongu Inn instead of Mongu
*This is now changed.*

Line 293: Which AERONET version was used? V3 came on line recently so it is not obvious.
*We have used version2/level 2 AERONET retrievals. This point is now detailled in the new version.*

Line 326: What was the altitude above clouds the reflectance was taken at?
*The reflectance measurement was taken at 1430 m, just prior to the profile. In-situ cloud data shows cloud top heights at 600m. In that sense, the reflectance has been measured about ~800 m above cloud tops. This specific point is now mentioned (Part 3.2.3.2).*

Line 554; CALIOP is all CAPS
*This is changed.*

Line 627: How much do you think assumptions of hygroscopicity versus speciation plays into this? Granted, this is mostly an absence of BC in smoke, but you still may have a factor of 2 floating around given the high RH in the MBL.
*We think the bias is mostly due to the aerosol speciation in this case. Hygroscopic properties that we used in the model is not able to explain such important differences in SSA. In addition, an external mixing hypothesis is used in the model, excluding the possible « lensing » effect (associated to increase of absorption).*

Line 635: can you elaborate here?
*This point is now more detailled in the new version.*

Paragraph around Line 683: This paragraph is bordering on a non-sequitur. Wv is a great tracer, but is fundamentally different from RH. So careful how you talk about humidity and optical properties.
*This is effectively right and we have now modified this paragraph.*

Paragraph around 695: Can you compare model versus measured f(RH) directly here?
*This point is a very interesting but, unfortunately, we could not conserve in the model both the dry and wet optical properties of smoke aerosols and we only calculate the wet properties for direct comparisons with in-situ/satellite observations.*

Paragraph starting 760:

On PBL impact: Please be specific what you mean by PBL height, are you referring to the actual top of the PBL (that can be somewhat amorphous given the depth of the entertainment zone in cloud atmospheres, but comes out often as a hazy model metric) or are you referring specifically to the top of the mixed layer? If you are referring to a systematic change in the base of the inversion, please state that clearly throughout. Also, Just curious, any wind impacts ? Mark Jacobson years ago was reporting big wind impacts in global GATOR simulations. See any evidence ? Regardless positive or negative it is worth mentioning any notable wind impacts.

*In this study, the PBL height corresponds to the top of the PBL. This is now detailed in the text. In parallel, we have also investigated the impact of BBA on the near-surface (10m) wind speed. The results are presented in the following figure. In our case, we obtain a general decrease (of about -0.5 m.s⁻¹) of the surface wind over most of the continent. Over the ocean, the impact of BBA is more complex with the presence of a regional contrast, characterized by an increase (decrease) of the surface wind around 0-10°W/15-30°S (latitudes higher than 15°S). This point is now mentioned in the part 5.2 of the new version.*

[Figure]

*Figure indicating the changes in the surface wind speed due to BBA (averaged for September 2016).*

*Also, can you calculate a specific surface temperature change per unit optical depth? See this for comparison https://www.atmos-chem-phys.net/16/6475/2016/*

*This is an interesting remark and we have now calculated the changes of surface temperature per unit of AOD due to the presence of BBA. The results we obtained (averaged for all the period of simulation) is about -2.5° per unit AOD (at 550 nm). We observe that this value is consistent and higher to the one (-1.5°) published by Zhang et al. (2016) for a massive biomass burning event occuring over Central Canada during June 2015. The difference could be due to the absorbing properties of BBA, which are more pronounced in the present study compared to Zhang et al. (2016) (SSA of 0.94). This could favor higher dimming effect and impact on the surface temperature over the Angola region. This interesting point is now mentioned in the part 5.2 and the reference of Zhang et al. (2016) is added.*

---

## Author Response (AR2)

Dear Authors

*Dear Editor,*

*Thank you for those comments we used to improve the manuscript,*

Congratulations on a much-improved draft. In general, my sense is that you have adequately addressed all the reviewer concerns. However, I think a "little bit" got through the gaps, so please take another read through the manuscript to make sure that the whole is harmonized for usage, corrections, etc. For one example, the correction of "Mongu Inn" was not extended throughout the manuscript. To me, this is not a significant issue, but since the Reviewer pointed it out, please either correct throughout (also the usage of "Mongu Station"), or add a sentence at the first use indicating that a variety of terms are used to reference this site.

*We have read the article to harmonize it. In parallel, the Mongu AERONET site is now mentioned as « Mongu Inn » throughout the manuscript,*

I also wanted to point out a more significant issue: in addressing Reviewer #1 on the difference between DRE and DRF, it appears that you have mixed them up by writing "(DRF; with respect to noaerosols)". This is of course a point that could be critical to the reader's understanding of everything going on, so please make certain that this is corrected throughout the manuscript. I refer you to Heald et al.'s paper on the topic: https://www.atmos-chem-phys.net/14/5513/2014/acp-14-5513-2014.html

*This is right and we have now replaced the term DRF by DRE over the whole manuscript following the Heald et al. (2014) convention as we have effectively investigated the « instantaneous radiative impact » of smoke aerosols in this study. In order to be precise, we have also included in the introduction the Heald et al. (2014) reference.*

[revised manuscript text omitted]